# Langevin Autoencoders for Learning Deep Latent Variable Models

## Abstract

Markov chain Monte Carlo (MCMC), such as Langevin dynamics, is valid for approximating intractable distributions. However, its usage is limited in the context of deep latent variable models since it is not scalable to data size owing to its datapoint-wise iterations and slow convergence. This paper proposes the *amortized Langevin dynamics* (ALD), wherein datapoint-wise MCMC iterations are entirely replaced with updates of an inference model that maps observations into latent variables. Since it no longer depends on datapoint-wise iterations, ALD enables scalable inference from large-scale datasets. Despite its efficiency, it retains the excellent property of MCMC; we prove that ALD has the target posterior as a stationary distribution under some assumptions. Furthermore, ALD can be extended to sampling from an unconditional distribution such as an energy-based model, enabling more flexible generative modeling by applying it to the prior distribution of the latent variable. Based on ALD, we construct a new deep latent variable model named the *Langevin autoencoder* (LAE). LAE uses ALD for autoencoder-like posterior inference and sampling from the latent space EBM. Using toy datasets, we empirically validate that ALD can properly obtain samples from target distributions in both conditional and unconditional cases, and ALD converges significantly faster than traditional LD. We also evaluate LAE on the image generation task using three datasets (SVHN, CIFAR-10, and CelebA-HQ). Not only can LAE be trained faster than non-amortized MCMC methods, but LAE can also generate better samples in terms of the Fréchet Inception Distance (FID) compared to AVI-based methods, such as the variational autoencoder[1].

## 1 Introduction

Variational inference (VI) and Markov chain Monte Carlo (MCMC) are two practical tools to approximate intractable distributions. Recently, VI has been dominantly used in deep latent variable models (DLVMs) to approximate the posterior distribution over the latent variable $\mathbf{z}$ given the observation $\mathbf{x}$, i.e., $p(\mathbf{z} \mid \mathbf{x})$. At the core of the success of VI is the invention of amortized variational inference (AVI) (Kingma et al., 2014; An & Cho, 2015; Su et al., 2018; Eslami et al., 2018; Kumar et al., 2018), which replaces optimization of datapoint-wise variational parameters with an inference model that predicts latent variables from observations. The advantage of AVI over traditional VI is that minibatch training can be used for its optimization, which enables efficient posterior inference on large-scale datasets. In addition, we can also leverage the optimized inference model to perform inference for new data in test time. However, the approximation power of AVI (or VI itself) is limited because it relies on distributions with tractable densities for approximations. Although there have been attempts to improve their flexibility (e.g., normalizing flows (Rezende & Mohamed, 2015; Kingma et al., 2016; Van Den Berg et al., 2018; Huang et al., 2018)), such methods typically have architectural constraints (e.g., invertibility in normalizing flows).

Compared to VI, MCMC can approximate complex distributions because it does not rely on any tractable distributions. Instead, MCMC repeats sampling from the target distribution and uses obtained samples to approximate the posterior distribution. Langevin dynamics is a typical example of MCMC for sampling from a continuous distribution. However, despite its high approximation ability, MCMC has received relatively little attention in learning DLVMs. It is because MCMC methods

---

[1]The implementation is available at `https://bit.ly/2Swow0F`

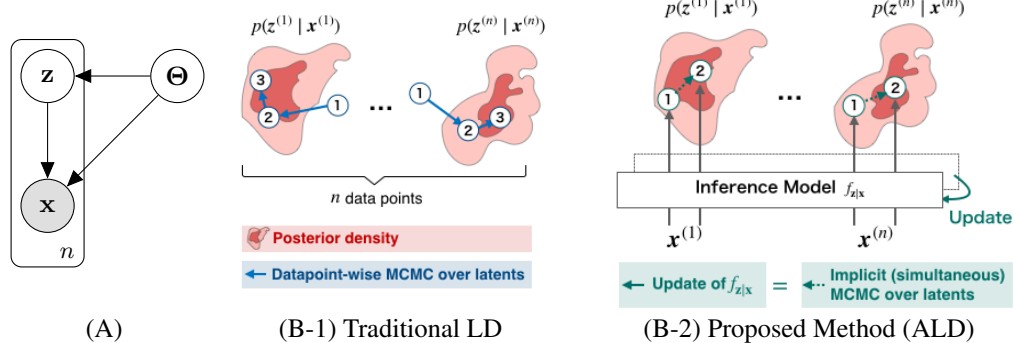

Figure 1: (A) Directed graphical model under consideration. (B-1) In traditional Langevin dynamics, the samples are directly updated in the latent space. (B-2) Our amortized Langevin dynamics replace the update of latent samples with an inference model $f_{\mathbf{z}|\mathbf{x}}$ that maps the observation $\boldsymbol{x}$ into the latent variable $\boldsymbol{z}$.

take a long time to converge, making it difficult to be used in the training of DLVMs. When learning DLVMs with MCMC, we need to run MCMC iterations for sampling from each posterior per data point, i.e., $p\left(\mathbf{z} \mid \boldsymbol{x}^{(i)}\right)$ $(i = 1, \ldots, n)$, where $n$ is the number of training data, as shown in Figure 1 (B-1). It is problematic, mainly when training with a large-scale $(n > 10\mathrm{K})$ dataset, because it is time-consuming to run massive MCMC iterations for all data points. Furthermore, we need to re-run the sampling procedure when we obtain new observations in test time.

As in VI, there have been some attempts to introduce the concept of amortized inference to MCMC. For example, Hoffman (2017) initializes MCMC sampling using an inference model that predicts latent variables from observations. However, as they use inference models only for the initialization of MCMC, these methods still rely on datapoint-wise sampling iterations. Not only is it time-consuming, but implementations of such partially amortized methods also tend to be complicated compared to the simplicity of AVI. To make MCMC more suitable for the inference of DLVMs, a more straightforward and sophisticated framework of amortization is needed.

This paper proposes the *amortized Langevin dynamics* (ALD), which replace datapoint-wise MCMC iterations with updates of an inference model that maps observations into latent variables (Figure 1 (B-2)). Since latent variables depend on the inference model, the updates of the inference model can be regarded as implicit updates of latent variables, which enables us to perform posterior inference without datapoint-wise MCMC iterations. Notably, our ALD treats outputs of the inference model themselves as samples from the target distribution, whereas existing amortization methods use the outputs only as initialization of MCMC. Therefore, ALD can be implemented straightforwardly like AVI. Moreover, despite the simplicity, we can theoretically guarantee that ALD has the true posterior as a stationary distribution under some assumptions, which is a critical requirement for valid MCMC algorithms.

Although we have introduced ALD as a posterior sampling algorithm, the application of ALD is not limited to sampling from posterior distributions. Recent studies have demonstrated that applying an energy-based model (EBM) into the prior distribution over the latent variable enables more flexible generative modeling for DLVMs. When we train an EBM, we have to obtain samples from the EBM by running costly MCMC iterations. By extending our ALD to sampling from unconditional distributions, we can apply ALD into sampling from such EBMs. When sampling from an EBM with ALD, we prepare a function that maps fixed inputs into latent variables, and updates of EBM samples are replaced with updates of the function's parameters. In the same way with the posterior case, the updates of the sampler function can be regarded as implicit updates of samples from EBMs.

Using our ALD for sampling from both the posterior and the EBM over the latent variable, we derive a novel framework of learning DLVMs, which we refer to as the *Langevin autoencoder* (LAE). Interestingly, the learning algorithm of LAE naturally takes the combined form of an autoencoder-like architecture and adversarial training. Our experiments show that ALD can properly obtain samples from target distributions using toy datasets. Subsequently, we perform numerical experiments of the image generation task using the SVHN, CIFAR-10, and CelebA-HQ datasets. Not only can LAE

be trained faster than non-amortized MCMC methods, but LAE can also generate better samples in terms of the Fréchet Inception Distance compared to AVI-based methods, such as VAE.

## 2 PRELIMINARIES

### 2.1 PROBLEM DEFINITION

Consider a probabilistic model with the $d_{\mathbf{x}}$-dimensional observation $\mathbf{x}$, the $d_{\mathbf{z}}$-dimensional continuous latent variable $\mathbf{z}$, and the model parameter $\boldsymbol{\Theta}$, as described by the probabilistic graphical model shown in Figure 1 (A). Although the posterior distribution over the latent variable is proportional to the prior and the likelihood: $p(\mathbf{z} \mid \mathbf{x}) = p(\mathbf{z}) p(\mathbf{x} \mid \mathbf{z}) / p(\mathbf{x})$, this is intractable due to the normalizing constant $p(\mathbf{x}) = \int p(\boldsymbol{z}) p(\mathbf{x} \mid \boldsymbol{z}) d\boldsymbol{z}$. This study aims to approximate the posterior $p(\mathbf{z} \mid \mathbf{x})$ for all $n$ observations $\boldsymbol{x}^{(1)}, \ldots \boldsymbol{x}^{(n)}$ efficiently by obtaining samples from it.

### 2.2 LANGEVIN DYNAMICS

Langevin dynamics (LD) (Neal, 2011) is a sampling algorithm based on the following Langevin equation:

$$d\boldsymbol{z} = -\nabla_{\boldsymbol{z}} U(\boldsymbol{x}, \boldsymbol{z}) \, dt + \sqrt{2\beta^{-1}} dB, \tag{1}$$

where $U$ is a Lipschitz continuous potential function that satisfies an appropriate growth condition, $\beta$ is an inverse temperature parameter, and $B$ is a Brownian motion. This stochastic differential equation has $p^{\beta}(\boldsymbol{z} \mid \boldsymbol{x}) \propto \exp(-\beta U(\boldsymbol{x}, \boldsymbol{z}))$ as its equilibrium distribution. We set $\beta = 1$ and define the potential as follows to obtain the target posterior $p(\mathbf{z} \mid \boldsymbol{x})$ as its equilibrium:

$$U(\boldsymbol{x}, \boldsymbol{z}) = -\log p(\boldsymbol{z}) - \log p(\boldsymbol{x} \mid \boldsymbol{z}). \tag{2}$$

We can obtain samples from the posterior by simulating Eq. (1) using the Euler–Maruyama method (Kloeden & Platen, 2013) as follows:

$$\boldsymbol{z}' \sim \mathcal{N}(\boldsymbol{z}'; \boldsymbol{z} - \eta \nabla_z U(\boldsymbol{x}, \boldsymbol{z}), 2\eta \boldsymbol{I}), \tag{3}$$

where $\eta$ is a step size for discretization. When the step size is sufficiently small, the samples asymptotically move to the target posterior by repeating this sampling iteration. LD can be applied to any posterior inference problems for continuous latent variables, provided the potential energy is differentiable on the latent space. However, to obtain the posterior samples for all observations $\boldsymbol{x}^{(1)}, \ldots \boldsymbol{x}^{(n)}$, we should perform iterations of Eq. (3) per data point, as shown in Figure 1 (B-1). It is inefficient, mainly if the dataset is large. In addition, we need to re-run the time-consuming iterations for new observations in test time. In the next section, we demonstrate a method that addresses the inefficiency by amortization.

## 3 AMORTIZED LANGEVIN DYNAMICS

### 3.1 GENERAL IDEA

As an alternative to the direct simulation of latent dynamics, we define an inference model $f_{\mathbf{z}|\mathbf{x}}$, which maps the observation into the latent variable. Formally, the dynamics of its parameter $\boldsymbol{\Phi}$ is

$$d\boldsymbol{\Phi} = -\sum_{i=1}^{n} \nabla_{\boldsymbol{\Phi}} U\left(\boldsymbol{x}^{(i)}, f_{\mathbf{z}|\mathbf{x}}\left(\boldsymbol{x}^{(i)}; \boldsymbol{\Phi}\right)\right) dt + \sqrt{2} dB. \tag{4}$$

Because the function $f_{\mathbf{z}|\mathbf{x}}$ outputs latent variables, the stochastic dynamics on the parameter space induce dynamics on the latent space. The main idea of our amortized Langevin dynamics (ALD) is to regard the transition on this induced dynamics as a sampling procedure for the posterior distributions, as shown in Figure 1 (B-2).

We can use the Euler–Maruyama method to simulate Eq. (4) like traditional LD:

$$\boldsymbol{\Phi}' \sim \mathcal{N}\left(\boldsymbol{\Phi}'; \boldsymbol{\Phi} - \eta \sum_{i=1}^{n} \nabla_{\boldsymbol{\Phi}} U\left(\boldsymbol{x}^{(i)}, f_{\mathbf{z}|\mathbf{x}}\left(\boldsymbol{x}^{(i)}; \boldsymbol{\Phi}\right)\right), 2\eta \boldsymbol{I}\right). \tag{5}$$

---

**Algorithm 1** Amortized Langevin dynamics

---

$\boldsymbol{\Phi} \leftarrow$ Initialize parameters
$\mathbb{Z}^{(1)}, \ldots, \mathbb{Z}^{(n)} \leftarrow \varnothing$ ▷ Initialize sample sets for all $n$ data points
**repeat**
    $\boldsymbol{\Phi} \leftarrow \boldsymbol{\Phi}' \sim \mathcal{N}\left(\boldsymbol{\Phi}'; \boldsymbol{\Phi} - \eta \sum_{i=1}^{n} \nabla_{\boldsymbol{\Phi}} U\left(\boldsymbol{x}^{(i)}, \boldsymbol{z}^{(i)} = f_{\mathbf{z}|\mathbf{x}}\left(\boldsymbol{x}^{(i)}; \boldsymbol{\Phi}\right)\right), 2\eta_\phi \boldsymbol{I}\right)$
    $\mathbb{Z}^{(1)}, \ldots, \mathbb{Z}^{(n)} \leftarrow \mathbb{Z}^{(1)} \cup \left\{f_{\mathbf{z}|\mathbf{x}}\left(\boldsymbol{x}^{(1)}; \boldsymbol{\Phi}\right)\right\}, \ldots, \mathbb{Z}^{(N)} \cup \left\{f_{\mathbf{z}|\mathbf{x}}\left(\boldsymbol{x}^{(n)}; \boldsymbol{\Phi}\right)\right\}$ ▷ Add samples
**until** convergence of parameters
**return** $\mathbb{Z}^{(1)}, \ldots, \mathbb{Z}^{(n)}$

---

Through the iterations, the posterior sampling is implicitly performed by collecting outputs of the inference model for each data point as described in Algorithm 1. Note that $\mathbb{Z}^{(i)}$ denotes a set of samples of the posterior for the $i$-th data (i.e., $p\left(\mathbf{z} \mid \boldsymbol{x}^{(i)}\right)$) obtained using ALD. When we perform inference for new test data, the trained inference model can be used to initialize an MCMC method (e.g., traditional LD) because it is expected that the trained inference model can map data into the high-density area of the posteriors.

By this amortization, we replace the direct update of latent variables $(\boldsymbol{z}^{(1)}, \ldots, \boldsymbol{z}^{(n)})$ with the update of the global parameter $\boldsymbol{\Phi}$. A significant advantage of amortization is that the cost of MCMC can be reduced by using minibatch training. For minibatch training, we substitute the minibatch statistics of $m$ data points for the derivative for all $n$ data.

$$\sum_{i=1}^{n} \nabla_{\boldsymbol{\Phi}} U\left(\boldsymbol{x}^{(i)}, f_{\mathbf{z}|\mathbf{x}}\left(\boldsymbol{x}^{(i)}; \boldsymbol{\Phi}\right)\right) \approx \frac{n}{m} \sum_{i=1}^{m} \nabla_\phi U\left(\boldsymbol{x}^{(i)}, f_{\mathbf{z}|\mathbf{x}}\left(\boldsymbol{x}^{(i)}\right)\right).$$

We refer to the minibatch version of ALD as *stochastic gradient amortized Langevin dynamics* (SGALD). SGALD enables us to sample from posteriors of a massive dataset efficiently. Moreover, in the context of stochastic gradient LD (SGLD), it is known that adaptive preconditioning effectively improves convergence compared to the naive SGLD (Li et al., 2016)[2]. This preconditioning technique is also applicable to our SGALD, and we employ it throughout our experiments.

### 3.2 THEORETICAL ANALYSIS

To justify our ALD as a posterior sampling algorithm, we provide a theoretical analysis of the stationary distribution of our ALD algorithm. Here, $\boldsymbol{X}$ and $\boldsymbol{Z}$ denote matrices with $\boldsymbol{x}^{(i)}$ and $\boldsymbol{z}^{(i)}$ in rows $\boldsymbol{X}_{i,:}$ and $\boldsymbol{Z}_{i,:}$, respectively. Our main result is as follows:

**Theorem 1.** *Let $q\left(\boldsymbol{Z} \mid \boldsymbol{X}\right)$ be a stationary distribution of the latent variables induced by Eq. (4). When the mapping $f_{\mathbf{z}|\mathbf{x}}$ meets the following conditions, $q\left(\boldsymbol{Z} \mid \boldsymbol{X}\right)$ satisfies $q\left(\boldsymbol{Z} \mid \boldsymbol{X}\right) \propto \exp\left(-U\left(\boldsymbol{X}, \boldsymbol{Z}\right)\right) \coloneqq \exp\left(-\sum_{i=1}^{n} U\left(\boldsymbol{x}^{(i)}, \boldsymbol{z}^{(i)}\right)\right).$*

1. *The mapping has the form of $f_{\mathbf{z}|\mathbf{x}}\left(\boldsymbol{x}; \boldsymbol{\Phi}\right) = \boldsymbol{\Phi} g\left(\boldsymbol{x}\right)$, where $\boldsymbol{\Phi}$ is a $d_{\mathbf{z}} \times d$ matrix, $g$ is a mapping from $\mathbb{R}^{d_{\mathbf{x}}}$ to $\mathbb{R}^d$, and $d$ is the dimensionality of $g\left(\boldsymbol{x}\right)$.*

2. *The rank of $\boldsymbol{G}$ is $n$, where $\boldsymbol{G}$ is a matrix with $g\left(\boldsymbol{x}^{(i)}\right)$ in row $\boldsymbol{G}_{i,:}$.*

See Appendix A for the proof. Theorem 1 suggests that samples obtained by ALD asymptotically converge to the true posterior when we construct the inference model $f_{\mathbf{z}|\mathbf{x}}$ with an appropriate form. Practically, we can implement such a function using a neural network whose parameters are fixed except for the last linear layer. In this implementation, the last linear layer takes a role of the parameter $\boldsymbol{\Phi}$, and the preceding feature extractor takes a role of the function $g$. In our experiments, we randomly initialize weights of the feature extractor and freeze them throughout the training.

In addition, the dimensionality of the last linear layer should be sufficiently large to meet the second condition. Therefore, the second condition derives a trade-off between approximation quality and

---

[2]The relationship between the naive SGLD and the preconditioned version is almost identical to the naive stochastic gradient descent and RMSProp.

computational costs. It is worth noting that a similar trade-off is also known in the context of AVI. In AVI, the approximation quality is influenced by the capacity of the inference model, and the gap between the optimal variational distribution and the amortized distribution is often denoted as the *amortization gap* (Cremer et al., 2018). In experiments, we confirm that preserving the condition does not become a significant computational overhead in practice. In addition, we should note that decaying the step size $\eta$ is needed to ensure convergence when we perform the simulation with discretization as described by Welling & Teh (2011).

### 3.3 EXTENSION TO UNCONDITIONAL CASES

Currently, we have introduced ALD as a sampling algorithm for conditional posterior distributions. We can also apply ALD to sampling from unconditional unnormalized distributions, namely energy-based models (EBMs). Consider an unconditional distribution over a random variable $\mathbf{z}$ defined by an energy function $f_{\mathbf{z}}$ as follows:

$$p\left(\boldsymbol{z}\right) \propto \exp\left(-f_{\mathbf{z}}\left(\boldsymbol{z}\right)\right), \tag{6}$$

where $f_{\mathbf{z}}$ maps the variable $\boldsymbol{z}$ into a scalar value. To obtain samples from this EBM using ALD, we prepare a sampler function $f_{\mathbf{z}|\mathbf{u}}\left(\boldsymbol{u};\boldsymbol{\Psi}\right)$ that maps its input $\boldsymbol{u}$ into the variable $\boldsymbol{z}$. Here, the input vector $\boldsymbol{u}$ is fixed, whereas observations are used in the posterior case. To run multiple MCMC chains in parallel, we prepare $k$ fixed inputs $\boldsymbol{u}^{(1)}, \ldots, \boldsymbol{u}^{(k)}$ and update the parameter of the sampler function as follows:

$$\boldsymbol{\Psi}' \sim \mathcal{N}\left(\boldsymbol{\Psi} - \eta \sum_{i=1}^{k} \nabla_{\boldsymbol{\Psi}} f_{\mathbf{z}}\left(f_{\mathbf{z}|\mathbf{u}}\left(\boldsymbol{u}^{(i)};\boldsymbol{\Psi}\right)\right), 2\eta\boldsymbol{I}\right). \tag{7}$$

Typically, the fixed input vectors $\boldsymbol{u}^{(1)}, \ldots, \boldsymbol{u}^{(k)}$ are chosen from a standard Gaussian distribution. As in the posterior case, we can guarantee the stationary distribution matches the EBM by choosing an appropriate form for the function $f_{\mathbf{z}|\mathbf{u}}$, and such a function can be implemented using a neural network whose parameters are fixed except for the last linear layer. For minibatch training, we can substitute the gradient for all $k$ chains with the stochastic gradient of $m$ minibatch chains:

$$\sum_{i=1}^{k} \nabla_{\boldsymbol{\Psi}} f_{\mathbf{z}}\left(f_{\mathbf{z}|\mathbf{u}}\left(\boldsymbol{u}^{(i)};\boldsymbol{\Psi}\right)\right) \approx \frac{k}{m} \sum_{i=1}^{m} \nabla_{\boldsymbol{\Psi}} f_{\mathbf{z}}\left(f_{\mathbf{z}|\mathbf{u}}\left(\boldsymbol{u}^{(i)};\boldsymbol{\Psi}\right)\right). \tag{8}$$

The advantage of using amortization in the unconditional case is that we can run massive chains parallel using minibatch training.

## 4 LANGEVIN AUTOENCODERS

Using ALD for sampling from both the posterior and the energy-based prior, we derive a novel framework for learning DLVMs. We here consider a latent variable model defined as follows:

$$p\left(\boldsymbol{z} \mid \boldsymbol{\Theta}\right) \propto \exp\left(-f_{\mathbf{z}}\left(\boldsymbol{z};\boldsymbol{\Theta}\right)\right), \; p\left(\boldsymbol{x} \mid \boldsymbol{z}, \boldsymbol{\Theta}\right) = \mathcal{N}\left(\boldsymbol{x}; f_{\mathbf{x}|\mathbf{z}}\left(\boldsymbol{z};\boldsymbol{\Theta}\right), \mathrm{diag}\left(\boldsymbol{\sigma}\right)\right), \tag{9}$$

where $\boldsymbol{\sigma}$ is a variance parameter of a Gaussian distribution[3]. To learn the latent variable model, we need to estimate both the model parameter $\boldsymbol{\Theta}$ and the latent variable $\mathbf{z}$. In Bayesian learning, the estimation is represented as the joint posterior distribution $p\left(\boldsymbol{\Theta}, \mathbf{z}^{(1)}, \ldots, \mathbf{z}^{(n)} \mid \boldsymbol{x}^{(1)}, \ldots, \boldsymbol{x}^{(n)}\right)$[4]. To obtain samples from the posterior, we can combine traditional LD and ALD as follows:

$$\boldsymbol{\Theta}' \sim \mathcal{N}\left(\boldsymbol{\Theta}'; \boldsymbol{\Theta} - \eta \nabla_{\boldsymbol{\Theta}} U\left(\boldsymbol{X}, f_{\mathbf{z}|\mathbf{x}}\left(\boldsymbol{X};\boldsymbol{\Phi}\right), \boldsymbol{\Theta}\right), 2\eta\boldsymbol{I}\right), \tag{10}$$

$$\boldsymbol{\Phi}' \sim \mathcal{N}\left(\boldsymbol{\Phi}'; \boldsymbol{\Phi} - \eta \nabla_{\boldsymbol{\Phi}} U\left(\boldsymbol{X}, f_{\mathbf{z}|\mathbf{x}}\left(\boldsymbol{X};\boldsymbol{\Phi}\right), \boldsymbol{\Theta}\right), 2\eta\boldsymbol{I}\right), \tag{11}$$

$$U\left(\boldsymbol{X}, \boldsymbol{Z}, \boldsymbol{\Theta}\right) := -\log p\left(\boldsymbol{\Theta}\right) - \sum_{i=1}^{n} \log p\left(\boldsymbol{z}^{(i)} \mid \boldsymbol{\Theta}\right) + \log p\left(\boldsymbol{x}^{(i)} \mid \boldsymbol{z}^{(i)}, \boldsymbol{\Theta}\right), \tag{12}$$

where $f_{\mathbf{z}|\mathbf{x}}\left(\boldsymbol{X};\boldsymbol{\Phi}\right)$ is a matrix with $f_{\mathbf{z}|\mathbf{x}}\left(\boldsymbol{x}^{(i)};\boldsymbol{\Phi}\right)$ in its $i$-th row. If we omit the Gaussian noise injection in Eq. (10), it corresponds to gradient ascent for maximum a posteriori (MAP) estimation

---

[3]We also include the variance $\boldsymbol{\sigma}$ into $\boldsymbol{\Theta}$ and treat it as a learnable parameter.
[4]We also provide the learning algorithm of maximum likelihood in Appendix B.

---

**Algorithm 2** Langevin Autoencoders

$\boldsymbol{\Theta}, \boldsymbol{\Phi}, \boldsymbol{\Psi} \leftarrow$ Initialize parameters
**repeat**
 $\boldsymbol{\Theta} \leftarrow \boldsymbol{\Theta}' \sim \mathcal{N}\left(\boldsymbol{\Theta}'; \boldsymbol{\Theta} - \eta \nabla_{\boldsymbol{\Theta}} \mathcal{L}\left(\boldsymbol{\Theta}, \boldsymbol{\Phi}, \boldsymbol{\Psi}\right), 2\eta \boldsymbol{I}\right)$      ▷ Update the generative model
 $\boldsymbol{\Phi} \leftarrow \boldsymbol{\Phi}' \sim \mathcal{N}\left(\boldsymbol{\Phi}'; \boldsymbol{\Phi} - \eta \nabla_{\boldsymbol{\Phi}} \mathcal{L}\left(\boldsymbol{\Theta}, \boldsymbol{\Phi}, \boldsymbol{\Psi}\right), 2\eta \boldsymbol{I}\right)$       ▷ Update the inference model
 $\boldsymbol{\Psi} \leftarrow \boldsymbol{\Psi}' \sim \mathcal{N}\left(\boldsymbol{\Psi}'; \boldsymbol{\Psi} + \eta \nabla_{\boldsymbol{\Psi}} \mathcal{L}\left(\boldsymbol{\Theta}, \boldsymbol{\Phi}, \boldsymbol{\Psi}\right), 2\eta \boldsymbol{I}\right)$       ▷ Update the sampler model
**until** convergence of parameters
**return** $\boldsymbol{\Theta}, \boldsymbol{\Phi}, \boldsymbol{\Psi}$

---

of $\boldsymbol{\Theta}$; if we additionally use a flat prior for $p\left(\boldsymbol{\Theta}\right)$, it yields the maximum likelihood estimation (MLE). In this study, we assume a flat prior for $p\left(\boldsymbol{\Theta}\right)$ and omit the notation for simplicity.

In Eq. (10), we cannot calculate the derivative of the potential function $\nabla_{\boldsymbol{\Theta}} U$ in a closed-form because the latent prior $p\left(\boldsymbol{z} \mid \boldsymbol{\Theta}\right)$ is defined using an unnormalized energy function. However, we can obtain the unbiased estimator of the derivative by obtaining samples from the prior as follows:

$$\nabla_{\boldsymbol{\Theta}} U\left(\boldsymbol{X}, \boldsymbol{Z}, \boldsymbol{\Theta}\right)$$

$$\approx \sum_{i=1}^{n} \nabla_{\boldsymbol{\Theta}} f_{\mathbf{z}}\left(\boldsymbol{z}^{(i)}; \boldsymbol{\Theta}\right) - \nabla_{\boldsymbol{\Theta}} \log p\left(\boldsymbol{x}^{(i)} \mid \boldsymbol{z}^{(i)}, \boldsymbol{\Theta}\right) - \frac{n}{k} \sum_{j=1}^{k} \nabla_{\boldsymbol{\Theta}} f_{\mathbf{z}}\left(\tilde{\boldsymbol{z}}^{(j)}; \boldsymbol{\Theta}\right), \quad (13)$$

where $\tilde{\boldsymbol{z}}^{(1)}, \ldots \tilde{\boldsymbol{z}}^{(n)}$ are sampled from the latent prior $p\left(\mathbf{z} \mid \boldsymbol{\Theta}\right)$ (see Appendix C for the derivation). To get samples from the latent prior, we can also use ALD by preparing a sampler function $f_{\mathbf{z}|\mathbf{u}}$ that takes fixed inputs $\boldsymbol{u}^{(1)}, \ldots, \boldsymbol{u}^{(k)}$, as described in Section 3.3. Here, we set the number of chains equal to the number of data points for simplicity, i.e., $k = n$.

In summary, the encoder $f_{\mathbf{z}|\mathbf{x}}$, the decoder $f_{\mathbf{x}|\mathbf{z}}$, and the latent energy function $f_{\mathbf{z}}$ are trained by minimizing the following loss function $\mathcal{L}$, whereas the latent sampler $f_{\mathbf{z}|\mathbf{u}}$ is trained by maximizing it, while stochastic noise of the Brownian motion is injected in their update in order to avoid shrinking to MAP estimates:

$$\mathcal{L}\left(\boldsymbol{\Theta}, \boldsymbol{\Phi}, \boldsymbol{\Psi}\right) \quad (14)$$

$$= \sum_{i=1}^{n} f_{\mathbf{z}}\left(f_{\mathbf{z}|\mathbf{x}}\left(\boldsymbol{x}^{(i)}; \boldsymbol{\Phi}\right); \boldsymbol{\Theta}\right) - f_{\mathbf{z}}\left(f_{\mathbf{z}|\mathbf{u}}\left(\boldsymbol{u}^{(i)}; \boldsymbol{\Psi}\right); \boldsymbol{\Theta}\right) - \log p\left(\boldsymbol{x}^{(i)} \mid f_{\mathbf{z}|\mathbf{x}}\left(\boldsymbol{x}^{(i)}; \boldsymbol{\Phi}\right), \boldsymbol{\Theta}\right).$$

We refer to this framework of learning DLVMs as the *Langevin autoencoder* (LAE). We summarize the algorithm of the LAE in Algorithm 2. LAE is closely related to the traditional autoencoder (AE) and other deep generative models, such as the variational autoencoder (VAE) and the generative adversarial network (GAN). We discuss the relationship in the next section in detail.

## 5 RELATED WORKS

**Amortized inference** is well-investigated in the context of variational inference. It is often referred to as *amortized variational inference* (AVI) (Rezende & Mohamed, 2015; Shu et al., 2018). The basic idea of AVI is to replace the optimization of the datapoint-wise variational parameters with the optimization of shared parameters across all data points by introducing an inference model that predicts latent variables from observations. The AVI is commonly used in generative models (Kingma & Welling, 2013), semi-supervised learning (Kingma et al., 2014), anomaly detection (An & Cho, 2015), machine translation (Su et al., 2018), and neural rendering (Eslami et al., 2018; Kumar et al., 2018). However, in the MCMC literature, there are few works on such amortization. Han et al. (2017) use traditional LD to obtain samples from posteriors to train deep latent variable models. Such Langevin-based algorithms for deep latent variable models are called *alternating back-propagation* (ABP) and are applied in several fields (Xie et al., 2019; Zhang et al., 2020; Xing et al., 2018; Zhu et al., 2019). However, ABP requires datapoint-wise Langevin iterations, causing slow convergence. Moreover, when we perform inference for new data in test time, ABP requires

MCMC iterations from randomly initialized samples again. Although Li et al. (2017) and Hoffman (2017) propose amortization methods for MCMC, they only amortize the initialization cost in MCMC by using an inference model. Therefore, they do not entirely remove datapoint-wise MCMC iterations.

**Autoencoders** (AEs) (Hinton & Salakhutdinov, 2006) can be seen as a particular case of LAEs, wherein the Gaussian noise injection to the update of the inference model (encoder) and the generative model (decoder) is omitted in Eqs. (10) and (11), and a flat prior is used for $p(\mathbf{z} \mid \mathbf{\Theta})$. When a different distribution is used as a latent prior, it is known as sparse autoencoders (SAEs) (Ng et al., 2011). In these cases, the dynamics in Eqs. (10) and (11) are dominated by gradient $\nabla U$; hence, both the latent variables and the model parameter converge to MLE or MAP estimates (or other stationary points). Therefore, AEs (and SAEs) can be considered MLE (and MAP) algorithms for the parameter $\mathbf{\Theta}$ and the latent variables $\mathbf{Z}$.

**Variational Autoencoders** (VAEs) are based on AVI, wherein an inference model (encoder) is defined as a variational distribution $q(\mathbf{z} \mid \mathbf{x}; \mathbf{\Phi})$ using a neural network. Its parameter $\mathbf{\Phi}$ is optimized by maximizing the evidence lower bound. Interestingly, there is a contrast between VAE and LAE when stochastic noise is used in posterior inference. In VAE, noise is used to sample from the stochastic inference model in calculating the potential $U$, i.e., in the *forward* calculation. However, in LAE, the inference model itself is deterministic, and stochastic noise is used for its parameter update along with the gradient calculation $\nabla_\phi U$, i.e., in the *backward* calculation. The advantage of LAE over VAE is that LAE can flexibly approximate complex posteriors by obtaining samples, whereas VAE's approximation ability is limited by choice of variational distribution $q(\mathbf{z} \mid \mathbf{x}; \mathbf{\Phi})$ because it requires a tractable density function. Although there are several considerations in the improvement of the approximation flexibility, these methods typically have architectural constraints (e.g., invertibility and ease of Jacobian calculation in normalizing flows (Rezende & Mohamed, 2015; Kingma et al., 2016; Van Den Berg et al., 2018; Huang et al., 2018; Titsias & Ruiz, 2019)), or they incur more computational costs (e.g., MCMC sampling for the reverse conditional distribution in unbiased implicit variational inference (Titsias & Ruiz, 2019)).

**Energy-based Models**' training is challenging, and many researchers have been studying methodology for its stable and practical training. A significant challenge is that it requires MCMC sampling from EBMs, which is challenging to perform in high dimensional space. Our LAE avoids this difficulty by defining the energy function in latent space rather than data space. A similar approach is taken in several works (Pang et al., 2020a;b), but they use traditional LD to obtain latent samples without amortization.

**Generative adversarial networks** (GANs) are closely related to our LAE because both are trained using adversarial loss functions. For a detailed discussion, see Appendix D.

## 6 EXPERIMENT

In our experiment, we first test our ALD algorithm on toy examples to investigate its behavior, then we show the results of its application to the training of deep generative models.

### 6.1 TOY EXAMPLES

We perform numerical simulation using toy examples to demonstrate that our ALD can properly obtain samples from target distributions in conditional and unconditional cases. First, we use examples where the posterior density can be derived in a closed-form. We initially generate three synthetic data $x_1, x_2, x_3$, where each $x_i$ is sampled from a bivariate Gaussian distribution as follows:

$$p(\boldsymbol{z}) = \mathcal{N}(\boldsymbol{z}; \boldsymbol{\mu}_{\mathbf{z}}, \boldsymbol{\Sigma}_{\mathbf{z}}), \quad p(\boldsymbol{x} \mid \boldsymbol{z}) = \mathcal{N}(\boldsymbol{x}; \boldsymbol{z}, \boldsymbol{\Sigma}_{\mathbf{x}}).$$

In this case, we can calculate the exact posterior as follows:

$$p(\boldsymbol{z} \mid \boldsymbol{x}) = \mathcal{N}\left(\boldsymbol{z}; \left(\boldsymbol{\Sigma}_{\mathbf{z}}^{-1} + \boldsymbol{\Sigma}_{\mathbf{x}}^{-1}\right)^{-1} \left(\boldsymbol{\Sigma}_{\mathbf{z}}^{-1} \boldsymbol{\mu}_{\mathbf{z}} + \boldsymbol{\Sigma}_{\mathbf{x}}^{-1} \boldsymbol{x}\right), \left(\boldsymbol{\Sigma}_{\mathbf{z}}^{-1} + \boldsymbol{\Sigma}_{\mathbf{x}}^{-1}\right)^{-1}\right),$$

In this experiment, we set $\boldsymbol{\mu}_{\mathbf{z}} = \begin{bmatrix} 0 \\ 0 \end{bmatrix}$, $\boldsymbol{\Sigma}_{\mathbf{z}} = \begin{bmatrix} 1 & 0 \\ 0 & 1 \end{bmatrix}$, and $\boldsymbol{\Sigma}_{\mathbf{x}} = \begin{bmatrix} 0.7 & 0.6 \\ 0.7 & 0.8 \end{bmatrix}$. We simulate our ALD algorithm for this setting to obtain samples from the posterior. We use a neural network

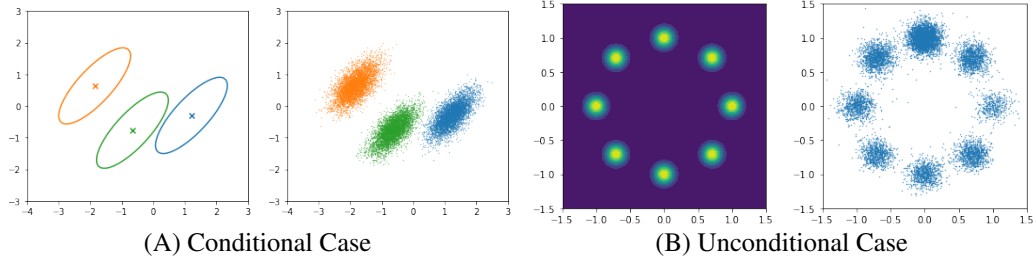

Figure 2: Visualization of ground truth density (left) and samples by ALD (right) in the conditional case (A) and the unconditional case (B) in toy examples.

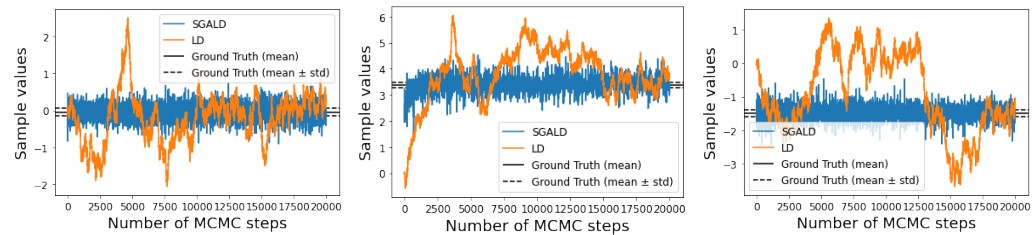

Figure 3: Evolution of sample values across MCMC iterations for traditional LD and our SGALD in univariate Gaussian examples. The black lines denote the ground truth posteriors (the solid lines show the mean values, and the dashed lines show the standard deviation).

of three fully connected layers of 128 units with ReLU activation for the inference model $f_{\mathbf{z}|\mathbf{x}}$; setting the step size to $4 \times 10^{-4}$, and update the parameters for 3,000 steps. We omit the first 1,000 samples as burn-in steps and use the remaining 2,000 samples for qualitative evaluation. The result is shown in Figure 2 (A). ALD produces samples that match the shape of the target distributions well, even though ALD does not perform direct updates of samples in the latent space. We also performed a similar experiment in a univariate setting to see the convergence speed of our SGALD, the minibatch version of ALD (see Appendix F.1 for the detailed experimental setting). Figure 3 shows the evolution of obtained sample values by traditional LD and our SGALD. It can be observed that SGALD's samples converge much faster than traditional LD.

In addition to the simple conjugate Gaussian example, we experiment with a complex posterior, wherein the likelihood is defined with a randomly initialized neural network. For comparison, we also implement the amortized variational inference (AVI) method, in which the posterior is approximated with a Gaussian distribution parameterized by a neural network (see Appendix F.2 for more experimental details). Figure 4 shows a typical example, which characterizes the difference between AVI and ALD. The advantage of our ALD over AVI is the flexibility of posterior approximation. AVI methods typically approximate posteriors using variational distributions, which have tractable density functions. Hence, their approximation power is limited by the choice of variational distribution family, and they often fail to approximate such complex posteriors, and they often fail to approximate such complex posteriors. On the other hand, ALD can capture such posteriors well. The results in other examples are summarized in Figure 6 in the appendix.

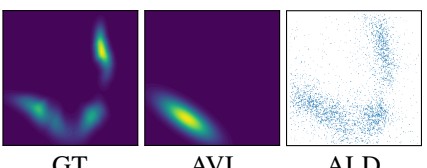

Figure 4: Visualizations of a ground truth posterior (left), an approximation by AVI (center), and samples by ALD (right) in the neural likelihood example.

Furthermore, we also test our ALD for sampling from an unconditional distribution. In this experiment, we use a mixture distribution of eight Gaussians and obtain samples using ALD, as shown in Figure 2 (B). We can observe that ALD adequately captures the actual density's multimodality and works well in the unconditional case.

Table 1: Quantitative results of the image generation for SVHN, CIFAR-10, and CelebA-HQ. We report the mean and standard deviation of the Fréchet Inception Distance in three different seeds.

| | Description | SVHN | CIFAR-10 | CelebA-HQ | |
| --- | --- | --- | --- | --- | --- |
| | | $32 \times 32$ | $32 \times 32$ | $32 \times 32$ | $64 \times 64$ |
| VAE | VI + amortization | $47.19 \pm 0.96$ | $106.0 \pm 0.5$ | $102.6 \pm 1.4$ | $174.9 \pm 0.9$ |
| VAE-flow | VI + amortization + flow | $46.69 \pm 0.69$ | $105.5 \pm 1.0$ | $101.2 \pm 1.1$ | $174.5 \pm 0.8$ |
| ABP | LD | $46.90 \pm 1.07$ | $105.6 \pm 0.2$ | $99.27 \pm 2.42$ | $135.7 \pm 2.0$ |
| DLGM | LD + amortized init. | $46.86 \pm 1.04$ | $102.3 \pm 1.76$ | $73.64 \pm 1.81$ | $139.9 \pm 3.4$ |
| LEBM | LD + EBM | $\mathbf{38.79} \pm 2.48$ | $97.02 \pm 0.36$ | $\mathbf{32.59} \pm 0.30$ | $\mathbf{53.31} \pm 2.26$ |
| LAE | LD + EBM + amortization | $46.66 \pm 1.33$ | $\mathbf{95.85} \pm 1.06$ | $40.33 \pm 1.33$ | $61.38 \pm 1.20$ |

## 6.2 IMAGE GENERATION

To demonstrate the applicability of our LAE to the generative model training, we experiment on image generation tasks using SVHN, CIFAR10, and CelebA-HQ datasets. Note that our goal here is not to provide the state-of-the-art results on image generation benchmarks but to verify the effectiveness of our ALD as a method of approximate inference in deep latent variable models. For this aim, we compare our LAE with five baseline methods, as shown in Table 1. VAE (Kingma & Welling, 2013) is one of the most popular deep latent variable models in which the posterior distribution is approximated using the AVI. VAE-flow is an extension of VAE in which the flexibility of AVI is improved using normalizing flows. In addition to AVI-based methods, we use three methods based on Langevin dynamics (LD). The alternating back-propagation (ABP) uses traditional LD to approximate the posterior, and the deep latent Gaussian model (DLGM) uses a VAE-like inference model to initialize LD. The latent energy-based model (LEBM) uses an EBM for the latent prior, and the EBM and posterior sampling is performed via traditional LD. LEBM can be regarded as a non-amortization version of our LAE.

We apply a commonly used convolutional neural network-based architecture for all models and a multi-layer perceptron for an energy-based model in the latent space of LAE and LEBM. Please refer to Appendix F.3 for more detailed experimental settings. For quantitative evaluation of the sample quality, we report the Fréchet Inception Distance (FID) (Heusel et al., 2017).

The results are summarized in Table 1. It can be observed that LAE outperforms VI-based methods in terms of FID, although it does not reach LEBM's performance in SVHN and CelebA-HQ. In training speed, LAE takes 34.12 seconds per epoch on average to train with CIFAR-10, while LEBM and DLGM take 84.29 and 60.66 seconds per epoch, respectively. This result shows that LAE is approximately 2.47 times faster than the non-amortized LEBM and 1.78 times faster than the partially amortized DLGM.

## 7 CONCLUSION

This paper proposed amortized Langevin dynamics (ALD), an efficient MCMC method for deep latent variable models. The ALD amortizes the cost of datapoint-wise iterations by using inference models. We showed that our ALD algorithm could accurately approximate posteriors with both theoretical and empirical studies. Using ALD, we derived a novel scheme of deep generative models called the *Langevin autoencoder* (LAE). We demonstrated that our LAE performs better than VI-based methods in sample quality and can be trained faster than non-amortized LD methods.

This study will be the first step to further work on efficient MCMC for latent variable models with large-scale datasets. For instance, deriving a Metropolis-Hastings rejection step for ALD and algorithms based on Hamiltonian Monte Carlo methods is an exciting direction of future work. Moreover, developing more sophisticated way of choosing the feature extractor of the inference model is also important. In our experiments, we used a randomly initialized neural network that is fixed throughout the training, but there could be a better way to improve the performance of LAE.

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

## A    PROOF OF THEOREM 1

First, we prepare some lemmas.

**Lemma 2.** *Let $h : \mathbb{R}^{d_{\mathbf{z}} \times d} \to \mathbb{R}^{d_{\mathbf{z}} \times n}$ be a linear map defined by $h(\mathbf{\Phi}) := \mathbf{\Phi} \mathbf{G}$. When the rank of $\mathbf{G}$ is $n$, there exists an orthogonal linear map $\tau : \mathbb{R}^{d_{\mathbf{z}} \times d} \to \mathbb{R}^{d_{\mathbf{z}} \times d}$ such that $\tau(\mathbf{\Phi}) = [\tilde{\mathbf{\Phi}}^{(1)}, \tilde{\mathbf{\Phi}}^{(2)}]$ satisfies $\ker \tilde{h} = \operatorname{span}\left[\mathbf{0}_{d_{\mathbf{z}} \times n}, \tilde{\mathbf{\Phi}}^{(2)}\right]$, where $\tilde{h} := h \circ \tau^{-1}$, $\tilde{\mathbf{\Phi}}^{(1)} := \left[\tilde{\phi}_1, \ldots, \tilde{\phi}_n\right]$, $\tilde{\mathbf{\Phi}}^{(2)} := \left[\tilde{\phi}_{n+1}, \ldots \tilde{\phi}_d\right]$ and $\tilde{\phi}_i \in \mathbb{R}^{d_{\mathbf{z}}}$ for $i = 1, \ldots, d$.*

*Proof.* The singular value decomposition of $\mathbf{G}$ is represented by $\mathbf{T} \mathbf{G} \tilde{\mathbf{T}}^\top = \begin{bmatrix} \mathbf{\Lambda} \\ \mathbf{0}_{(d-n \times n)} \end{bmatrix}$, where $\mathbf{\Lambda}$ is a $n \times n$ diagonal matrix $\mathbf{\Lambda} = \operatorname{diag}(\lambda_1, \ldots, \lambda_n)$, and $\mathbf{T}$ and $\tilde{\mathbf{T}}$ are orthogonal matrices. Since the rank of $\mathbf{G}$ is $n$, $\lambda_1, \ldots, \lambda_n$ are non-zero. When we set $\tau(\mathbf{\Phi}) := \mathbf{\Phi} \mathbf{T}^\top$, we obtain

$$
\begin{aligned}
\tilde{h}\left(\tilde{\mathbf{\Phi}}\right) &:= h\left(\tau^{-1}\left(\tilde{\mathbf{\Phi}}\right)\right) \\
&= \tilde{\mathbf{\Phi}} \mathbf{T} \mathbf{G} = \tilde{\mathbf{\Phi}} \left(\mathbf{T} \mathbf{G} \tilde{\mathbf{T}}^\top\right) \tilde{\mathbf{T}} \\
&= \tilde{\mathbf{\Phi}} \begin{bmatrix} \mathbf{\Lambda} \\ \mathbf{0}_{(d-n \times n)} \end{bmatrix} \tilde{\mathbf{T}}.
\end{aligned}
\tag{15}
$$

From the above equation, $\ker \tilde{h} = \operatorname{span}\left[\mathbf{0}_{d' \times n}, \tilde{\mathbf{\Phi}}^{(2)}\right]$ holds. $\square$

**Lemma 3.** *For $\tilde{\mathbf{\Phi}} \in \mathbb{R}^{d_{\mathbf{z}} \times d}$, let $\mathbf{\Phi}$ satisfy:*

$$
\tilde{\mathbf{\Phi}} = \left[\tilde{\mathbf{\Phi}}^{(1)}, \tilde{\mathbf{\Phi}}^{(2)}\right] = \tau(\mathbf{\Phi}).
$$

*A map $\tilde{h}^{(1)} : \mathbb{R}^{d_{\mathbf{z}} \times n} \to \mathbb{R}^{d_{\mathbf{z}} \times n}$ defined by $\tilde{h}^{(1)}\left(\tilde{\mathbf{\Phi}}^{(1)}\right) := \tilde{h}\left(\left[\tilde{\mathbf{\Phi}}^{(1)}, \mathbf{0}\right]\right)$ satisfies $\mathbf{\Phi} \mathbf{G} = \tilde{h}^{(1)}\left(\tilde{\mathbf{\Phi}}^{(1)}\right)$ and is linear isomorphic.*

*Proof.* From the definition, we have

$$
\begin{aligned}
\mathbf{\Phi} \mathbf{G} = \tau^{-1}\left(\tilde{\mathbf{\Phi}}\right) \mathbf{G} &= \tilde{\mathbf{\Phi}} \mathbf{T} \mathbf{G} \\
&= \tilde{\mathbf{\Phi}} \left(\mathbf{T} \mathbf{G} \tilde{\mathbf{T}}^\top\right) \tilde{\mathbf{T}} \\
&= \left[\tilde{\mathbf{\Phi}}^{(1)}, \tilde{\mathbf{\Phi}}^{(2)}\right] \begin{bmatrix} \mathbf{\Lambda} \\ \mathbf{0}_{(d-n \times n)} \end{bmatrix} \tilde{\mathbf{T}} \\
&= \left[\tilde{\mathbf{\Phi}}^{(1)}, \mathbf{0}\right] \begin{bmatrix} \mathbf{\Lambda} \\ \mathbf{0}_{(d-n \times n)} \end{bmatrix} \tilde{\mathbf{T}} \\
&= \tau^{-1}\left(\left[\tilde{\mathbf{\Phi}}^{(1)}, \mathbf{0}\right]\right) \mathbf{G} \\
&= h \circ \tau^{-1}\left(\left[\tilde{\mathbf{\Phi}}^{(1)}, \mathbf{0}\right]\right) = \tilde{h}\left(\left[\tilde{\mathbf{\Phi}}^{(1)}, \mathbf{0}\right]\right) \\
&= \tilde{h}^{(1)}\left(\tilde{\mathbf{\Phi}}^{(1)}\right).
\end{aligned}
$$

By the definition, $\tilde{h}^{(1)}$ is linear. Here, $\tilde{h}^{(1)}$ is injective, since $\ker \tilde{h} = \operatorname{span}\left[\mathbf{0}_{d' \times n}, \tilde{\mathbf{\Phi}}^{(2)}\right]$, and hence, $\dim\left(\operatorname{Im} \tilde{h}^{(1)}\right) \geq d_{\mathbf{z}} \times n$. Since $\operatorname{Im} \tilde{h}^{(1)} \subset \mathbb{R}^{d_{\mathbf{z}} \times n}$, $\tilde{h}^{(1)}$ is surjective. $\square$

**Lemma 4.** *For $V : \mathbb{R}^D \ni \phi \mapsto V(\mathbf{\Phi}) := U\left(\mathbf{X}, f_{\mathbf{z}|\mathbf{x}}(\mathbf{X}; \mathbf{\Phi})\right) \in \mathbb{R}$, $\tilde{\mathbf{\Phi}} = \left[\tilde{\mathbf{\Phi}}^{(1)}, \tilde{\mathbf{\Phi}}^{(2)}\right] := \tau(\mathbf{\Phi})$, $\tilde{V} := V \circ \tau^{-1}$ and $\tilde{V}^{(1)}\left(\tilde{\mathbf{\Phi}}^{(1)}\right) := \tilde{V}\left(\left[\tilde{\mathbf{\Phi}}^{(1)}, \mathbf{0}_{d_{\mathbf{z}} \times (d-n)}\right]\right)$, Eq. (4) is equivalent to*

$$
d\tilde{\mathbf{\Phi}}^{(1)} = -\nabla_{\tilde{\mathbf{\Phi}}^{(1)}} \tilde{V}^{(1)}\left(\tilde{\mathbf{\Phi}}^{(1)}\right) dt + \sqrt{2} dB,
\tag{16}
$$

$$
d\tilde{\mathbf{\Phi}}^{(2)} = \sqrt{2} dB.
\tag{17}
$$

*Proof.* By direct calculation, we obtain

$$
\begin{aligned}
\tilde{V}\left(\left[\tilde{\boldsymbol{\Phi}}^{(1)}, \tilde{\boldsymbol{\Phi}}^{(2)}\right]\right) &= V \circ \tau^{-1}\left(\left[\tilde{\boldsymbol{\Phi}}^{(1)}, \tilde{\boldsymbol{\Phi}}^{(2)}\right]\right) \\
&= U\left(\boldsymbol{X}, f_{\mathbf{z}|\mathbf{x}}\left(\boldsymbol{X}; \tau^{-1}\left(\left[\tilde{\boldsymbol{\Phi}}^{(1)}, \tilde{\boldsymbol{\Phi}}^{(2)}\right]\right)\right)\right) \\
&= U\left(\boldsymbol{X}, h\left(\tau^{-1}\left(\left[\tilde{\boldsymbol{\Phi}}^{(1)}, \tilde{\boldsymbol{\Phi}}^{(2)}\right]\right)\right)\right) \\
&= U\left(\boldsymbol{X}, h \circ \tau^{-1}\left(\left[\tilde{\boldsymbol{\Phi}}^{(1)}, \mathbf{0}\right]\right) + h \circ \tau^{-1}\left(\left[\mathbf{0}, \tilde{\boldsymbol{\Phi}}^{(2)}\right]\right)\right) \\
&= U\left(\boldsymbol{X}, h\left(\tau^{-1}\left(\left[\tilde{\boldsymbol{\Phi}}^{(1)}, \mathbf{0}\right]\right)\right)\right) \\
&= U\left(\boldsymbol{X}, f_{\mathbf{z}|\mathbf{x}}\left(\boldsymbol{X}; \tau^{-1}\left(\left[\tilde{\boldsymbol{\Phi}}^{(1)}, \mathbf{0}\right]\right)\right)\right) \\
&= V \circ \tau^{-1}\left(\left[\tilde{\boldsymbol{\Phi}}^{(1)}, \mathbf{0}\right]\right) \\
&= \tilde{V}\left(\left[\tilde{\boldsymbol{\Phi}}^{(1)}, \mathbf{0}\right]\right).
\end{aligned} \tag{18}
$$

Then, the following equivalence holds:

$$
\begin{aligned}
d\boldsymbol{\Phi} &= -\nabla_{\boldsymbol{\Phi}} V\left(\boldsymbol{\Phi}\right) dt + \sqrt{2} dB, \\
\Leftrightarrow d\tau^{-1}\left(\boldsymbol{\Phi}\right) &= -\tau^{\top}\left(\nabla_{\tilde{\boldsymbol{\Phi}}} \tilde{V}\left(\tilde{\boldsymbol{\Phi}}\right)\right) dt + \sqrt{2} dB \\
\Leftrightarrow d\tilde{\boldsymbol{\Phi}} &= -\tau \circ \tau^{\top}\left(\nabla_{\tilde{\boldsymbol{\Phi}}} \tilde{V}\left(\tilde{\boldsymbol{\Phi}}\right)\right) dt + \sqrt{2} d\tau\left(B\right) \\
&= -\nabla_{\tilde{\boldsymbol{\Phi}}} \tilde{V}\left(\tilde{\boldsymbol{\Phi}}\right) dt + \sqrt{2} dB,
\end{aligned} \tag{19}
$$

where we used $\tau \circ \tau^{\top} = \text{id}$ because $\tau$ is orthogonal. From Eq. (18), the dynamics in Eq. (19) is equivalent to Eq. (16) and Eq. (17). $\qquad\square$

In the following, we prove Theorem 1 using the above lemmas. The latent variables $\boldsymbol{Z} := \tilde{h}^{(1)}\left(\tilde{\boldsymbol{\Phi}}^{(1)}\right)$ is independent of $\tilde{\boldsymbol{\Phi}}^{(2)}$, and the probability distribution $q\left(\boldsymbol{Z} \mid \boldsymbol{X}\right)$ of $\boldsymbol{Z}$ is given by the pushforward measure $\left(\tilde{h}^{(1)}\right)_{\#}\left(p_*^{(1)}\right)\left(\boldsymbol{Z}\right)$ of the probability distribution $p^{(1)}$ of $\tilde{\boldsymbol{\Phi}}$ by $\tilde{h}^{(1)}$. The amortized Langevin dynamics has $q\left(\boldsymbol{Z} \mid \boldsymbol{G}\right)$ as its stationary distribution of $\boldsymbol{Z}$. Then, we have

$$
\begin{aligned}
q\left(\boldsymbol{Z} \mid \boldsymbol{X}\right) &= \left(\tilde{h}^{(1)}\right)_{\#}\left(p_*^{(1)}\right)\left(\boldsymbol{Z}\right) \\
&= p^{(1)}\left(\left(\tilde{h}^{(1)}\right)^{-1}\left(\boldsymbol{Z}\right)\right)\left|\det \frac{d(\tilde{h}^{(1)})^{-1}}{d\boldsymbol{Z}}\right| \\
&= p^{(1)}\left(\left(\tilde{h}^{(1)}\right)^{-1}\left(\boldsymbol{Z}\right)\right)\left|\det \frac{d\tilde{h}^{(1)}}{d\tilde{\boldsymbol{\Phi}}^{(1)}}\right|^{-1} \\
&= p^{(1)}\left(\left(\tilde{h}^{(1)}\right)^{-1}\left(\boldsymbol{Z}\right)\right) \times \left|\det \tilde{h}^{(1)}\right|^{-1} \\
&\propto \exp\left(-\tilde{V}\left(\left(\tilde{h}^{(1)}\right)^{-1}\left(\boldsymbol{Z}\right)\right)\right) \\
&= \exp\left(-V\left(\tau^{-1}\left(\left[\left(\tilde{h}^{(1)}\right)^{-1}\left(\boldsymbol{Z}\right), \mathbf{0}\right]\right)\right)\right) \\
&= \exp\left(-U\left(\boldsymbol{X}, \tau^{-1}\left(\left[\left(\tilde{h}^{(1)}\right)^{-1}\left(\boldsymbol{Z}\right), \mathbf{0}\right]\right) \boldsymbol{G}\right)\right) \\
&= \exp\left(-U\left(\boldsymbol{X}, \boldsymbol{Z}\right)\right),
\end{aligned}
$$

where we used that $\frac{d\tilde{h}^{(1)}}{d\tilde{\boldsymbol{\Phi}}^{(1)}} = \tilde{h}^{(1)}$ because of the linearity of $\tilde{h}^{(1)}$ and is constant with respect to $\boldsymbol{Z}$. The last equation is derived as follows. From Lemma 3, $\boldsymbol{\Phi} \boldsymbol{G} = \tilde{h}^{(1)}\left(\tilde{\boldsymbol{\Phi}}^{(1)}\right)$ holds when

---

**Algorithm 3** Amortized Langevin dynamics (test time)

---

$\boldsymbol{z} \leftarrow f_{\mathbf{z}|\mathbf{x}}\left(\boldsymbol{x}; \boldsymbol{\Phi}^*\right)$        ▷ Initialize a sample using a trained inference model
$\mathbb{Z} \leftarrow \varnothing$        ▷ Initialize a sample set
**repeat**
    $\boldsymbol{z} \leftarrow \boldsymbol{z}' \sim \mathcal{N}\left(\boldsymbol{z}'; \boldsymbol{z} - \eta \nabla_z U\left(\boldsymbol{x}, \boldsymbol{z}\right), 2\eta \boldsymbol{I}\right)$        ▷ Update the sample using traditional LD
    $\mathbb{Z} \leftarrow \mathbb{Z} \cup \{\boldsymbol{z}\}$        ▷ Add samples
**until** convergence of parameters
**return** $\mathbb{Z}$

---

$\tilde{\boldsymbol{\Phi}} = \left[\tilde{\boldsymbol{\Phi}}^{(1)}, \tilde{\boldsymbol{\Phi}}^{(2)}\right] = \tau\left(\boldsymbol{\Phi}\right)$. Thus, when $\boldsymbol{\Phi} = \tau^{-1}\left(\left[\tilde{\boldsymbol{\Phi}}^{(1)}, \boldsymbol{0}\right]\right)$, we obtain $\tilde{h}^{(1)}\left(\tilde{\boldsymbol{\Phi}}^{(1)}\right) = \boldsymbol{\Phi}\boldsymbol{G} = \tau^{-1}\left(\left[\tilde{\boldsymbol{\Phi}}^{(1)}, \boldsymbol{0}\right]\right)\boldsymbol{G}$. In particular, for $\tilde{\boldsymbol{\Phi}}^{(1)} = \left(\tilde{h}^{(1)}\right)^{-1}\left(\boldsymbol{Z}\right)$, we have

$$\boldsymbol{Z} = \tilde{h}^{(1)}\left(\left(\tilde{h}^{(1)}\right)^{-1}\left(\boldsymbol{Z}\right)\right)$$
$$= \tau^{-1}\left(\left[\left(\tilde{h}^{(1)}\right)^{-1}\left(\boldsymbol{Z}\right), \boldsymbol{0}\right]\right)\boldsymbol{G}.$$

$\square$

## B    MAXIMUM LIKELIHOOD TRAINING FOR LAE

In Section 4, we derive Bayesian learning algorithm of LAE, where the whole model is formulated as a Bayesian neural network. We can also think of the frequentist approach, where the training is defined as maximum likelihood. In this case, the objective is to maximize the evidence $\log p\left(\boldsymbol{x} \mid \boldsymbol{\Theta}\right)$, and its derivative is as follows.

$$\nabla_{\boldsymbol{\Theta}} \log p\left(\boldsymbol{x} \mid \boldsymbol{\Theta}\right) \tag{20}$$

$$= \nabla_{\boldsymbol{\Theta}} \log \int p\left(\boldsymbol{x}, \boldsymbol{z} \mid \boldsymbol{\Theta}\right) d\boldsymbol{z} \tag{21}$$

$$= \frac{1}{p\left(\boldsymbol{x} \mid \boldsymbol{\Theta}\right)} \int \nabla_{\boldsymbol{\Theta}} p\left(\boldsymbol{x}, \boldsymbol{z} \mid \boldsymbol{\Theta}\right) d\boldsymbol{z} \tag{22}$$

$$= \int \frac{p\left(\boldsymbol{x}, \boldsymbol{z} \mid \boldsymbol{\Theta}\right)}{p\left(\boldsymbol{x} \mid \boldsymbol{\Theta}\right)} \nabla_{\boldsymbol{\Theta}} \log p\left(\boldsymbol{x}, \boldsymbol{z} \mid \boldsymbol{\Theta}\right) d\boldsymbol{z} \tag{23}$$

$$= \mathbb{E}_{p(\boldsymbol{z}|\boldsymbol{x},\boldsymbol{\Theta})}\left[\nabla_{\boldsymbol{\Theta}} \log p\left(\boldsymbol{x}, \boldsymbol{z} \mid \boldsymbol{\Theta}\right)\right] \tag{24}$$

$$= \mathbb{E}_{p(\boldsymbol{z}|\boldsymbol{x},\boldsymbol{\Theta})}\left[\nabla_{\boldsymbol{\Theta}} \log p\left(\boldsymbol{x} \mid \boldsymbol{z}, \boldsymbol{\Theta}\right) + \nabla_{\boldsymbol{\Theta}} \log p\left(\boldsymbol{z} \mid \boldsymbol{\Theta}\right)\right] \tag{25}$$

$$= \mathbb{E}_{p(\boldsymbol{z}|\boldsymbol{x},\boldsymbol{\Theta})}\left[\nabla_{\boldsymbol{\Theta}} \log p\left(\boldsymbol{x} \mid \boldsymbol{z}, \boldsymbol{\Theta}\right) - \nabla_{\boldsymbol{\Theta}} f_{\mathbf{z}}\left(\boldsymbol{z}; \boldsymbol{\Theta}\right)\right] + \mathbb{E}_{p(\boldsymbol{z}|\boldsymbol{\Theta})}\left[\nabla_{\boldsymbol{\Theta}} f_{\mathbf{z}}\left(\boldsymbol{z}; \boldsymbol{\Theta}\right)\right]. \tag{26}$$

We can approximate the derivative by obtaining samples from the posterior $p\left(\boldsymbol{z} \mid \boldsymbol{x}, \boldsymbol{\Theta}\right)$ and the latent prior $p\left(\boldsymbol{z} \mid \boldsymbol{\Theta}\right)$ using LAE as in Eq. (5) and (7). Hence, the maximum likelihood version of LAE is summarized as shown in Algorithm 4.

## C    DERIVATION OF EQ. (13)

$$\nabla_{\boldsymbol{\Theta}} U\left(\boldsymbol{X}, \boldsymbol{Z}, \boldsymbol{\Theta}\right)$$
$$= \sum_{i=1}^{n} \nabla_{\boldsymbol{\Theta}} f_{\mathbf{z}}\left(\boldsymbol{z}^{(i)}; \boldsymbol{\Theta}\right) - \nabla_{\boldsymbol{\Theta}} \log p\left(\boldsymbol{x}^{(i)} \mid \boldsymbol{z}^{(i)}, \boldsymbol{\Theta}\right) + \nabla_{\boldsymbol{\Theta}} \log C\left(\boldsymbol{\Theta}\right),$$

---

**Algorithm 4** Maximum likelihood version of Langevin Autoencoders

$\boldsymbol{\Theta}, \boldsymbol{\Phi}, \boldsymbol{\Psi} \leftarrow$ Initialize parameters
**repeat**
   **repeat**
      $\boldsymbol{\Phi} \leftarrow \boldsymbol{\Phi}' \sim \mathcal{N}\left(\boldsymbol{\Phi}'; \boldsymbol{\Phi} - \eta \nabla_{\boldsymbol{\Phi}} \mathcal{L}\left(\boldsymbol{\Theta}, \boldsymbol{\Phi}, \boldsymbol{\Psi}\right), 2\eta \boldsymbol{I}\right)$           $\triangleright$ Update the inference model
      $\boldsymbol{\Psi} \leftarrow \boldsymbol{\Psi}' \sim \mathcal{N}\left(\boldsymbol{\Psi}'; \boldsymbol{\Psi} + \eta \nabla_{\boldsymbol{\Psi}} \mathcal{L}\left(\boldsymbol{\Theta}, \boldsymbol{\Phi}, \boldsymbol{\Psi}\right), 2\eta \boldsymbol{I}\right)$           $\triangleright$ Update the sampler model
   **until** convergence of $\boldsymbol{\Phi}$ and $\boldsymbol{\Psi}$
   $\boldsymbol{\Theta} \leftarrow \boldsymbol{\Theta} - \eta \nabla_{\boldsymbol{\Theta}} \mathcal{L}\left(\boldsymbol{\Theta}, \boldsymbol{\Phi}, \boldsymbol{\Psi}\right)$           $\triangleright$ Update the generative model
**until** convergence of $\boldsymbol{\Theta}$
**return** $\boldsymbol{\Theta}, \boldsymbol{\Phi}, \boldsymbol{\Psi}$

---

where $C\left(\boldsymbol{\Theta}\right) = \int \exp\left(-f_{\mathbf{z}}\left(\boldsymbol{z}; \boldsymbol{\Theta}\right)\right) d\boldsymbol{z}$. By direct calculation, we obtain

$$\nabla_{\boldsymbol{\Theta}} \log C\left(\boldsymbol{\Theta}\right) = \frac{1}{C\left(\boldsymbol{\Theta}\right)} \nabla_{\boldsymbol{\Theta}} C\left(\boldsymbol{\Theta}\right) \tag{27}$$

$$= \frac{1}{C\left(\boldsymbol{\Theta}\right)} \int \nabla_{\boldsymbol{\Theta}} \exp\left(-f_{\mathbf{z}}\left(\boldsymbol{z}; \boldsymbol{\Theta}\right)\right) d\boldsymbol{z} \tag{28}$$

$$= -\int \frac{\exp\left(-f_{\mathbf{z}}\left(\boldsymbol{z}; \boldsymbol{\Theta}\right)\right)}{C\left(\boldsymbol{\Theta}\right)} \nabla_{\boldsymbol{\Theta}} f_{\mathbf{z}}\left(\boldsymbol{z}; \boldsymbol{\Theta}\right) d\boldsymbol{z} \tag{29}$$

$$= -\int p\left(\boldsymbol{z} \mid \boldsymbol{\Theta}\right) \nabla_{\boldsymbol{\Theta}} f_{\mathbf{z}}\left(\boldsymbol{z}; \boldsymbol{\Theta}\right) d\boldsymbol{z} \tag{30}$$

$$= -\mathbb{E}_{\boldsymbol{z} \sim p(\mathbf{z}|\boldsymbol{\Theta})} \left[\nabla_{\boldsymbol{\Theta}} f_{\mathbf{z}}\left(\boldsymbol{z}; \boldsymbol{\Theta}\right)\right] \tag{31}$$

$$\approx -\frac{1}{k} \sum_{j=1}^{k} \nabla_{\boldsymbol{\Theta}} f_{\mathbf{z}}\left(\tilde{\boldsymbol{z}}^{(j)}; \boldsymbol{\Theta}\right), \tag{32}$$

where $\tilde{\boldsymbol{z}}^{(1)}, \ldots \tilde{\boldsymbol{z}}^{(n)}$ are samples drawn from $p\left(\mathbf{z} \mid \boldsymbol{\Theta}\right)$.

## D ADDITIONAL RELATED WORK

**Generative Adversarial Network** (GAN) (Goodfellow et al., 2014) is similar to LAE in that both are trained by minimax game between two functions (i.e., the energy function and the sampler function in LAE; the discriminator and the generator in GAN). However, there are some differences between them. First, the minimax game is performed in the latent space in LAE, while it is performed in the observation space in GAN. In other words, the latent variable is identical to the observation (i.e., $p\left(\boldsymbol{x} \mid \boldsymbol{z}\right) = \mathbf{1}_{\boldsymbol{x}=\boldsymbol{z}}$) in GAN. Note that the latent variable $\boldsymbol{z}$ is different from the input of GAN's generators. Here, the input of the GAN's generators is denoted as $\boldsymbol{u}$ for the analogy with LAE. Second, the loss function is slightly different. In GAN, the loss function is as follows:

$$\mathcal{L}_{\text{GAN}}\left(\boldsymbol{\Theta}, \boldsymbol{\Psi}\right) = -\sum_{i=1}^{n} \log f_{\mathbf{x}}\left(\boldsymbol{x}^{(i)}; \boldsymbol{\Theta}\right) + \log\left(1 - f_{\mathbf{x}}\left(f_{\mathbf{x}|\mathbf{u}}\left(\boldsymbol{u}^{(i)}; \boldsymbol{\Psi}\right); \boldsymbol{\Theta}\right)\right), \tag{33}$$

where $f_{\mathbf{x}|\mathbf{u}}$ denotes the generator that maps its inputs $\boldsymbol{u}$ into the observation space, and $f_{\mathbf{x}}$ denotes the discriminator that maps from the observation space into $(0, 1)$, and $\boldsymbol{u}^{(i)} \sim \mathcal{N}\left(\boldsymbol{u}; \mathbf{0}, \boldsymbol{I}\right)$. The discriminator is trained to minimize this loss function, whereas the generator is trained to maximize it. The main difference to the loss function of LAE is the second term. When we substitute it with $-\log f_{\mathbf{x}}\left(f_{\mathbf{x}|\mathbf{u}}\left(\boldsymbol{u}^{(i)}; \boldsymbol{\Psi}\right); \boldsymbol{\Theta}\right)$, it becomes more similar. This modification is often used to alleviate gradient vanishing and stabilize the training of GAN's generator (Goodfellow et al., 2014; Johnson & Zhang, 2018). In this formulation, the counter parts of the energy function and the sampler function are $-\log f_{\mathbf{x}}\left(\cdot; \boldsymbol{\Theta}\right)$ and $f_{\mathbf{x}|\mathbf{u}}\left(\cdot; \boldsymbol{\Psi}\right)$, respectively.

Another difference between LAE and GAN is that the input vector of the sampler function is fixed through the training in LAE, whereas the input of the generator changes per iteration by sampling from $\mathcal{N}\left(\boldsymbol{u}; \mathbf{0}, \boldsymbol{I}\right)$ in GAN. Furthermore, LAE is trained using noise injected gradient, whereas GAN

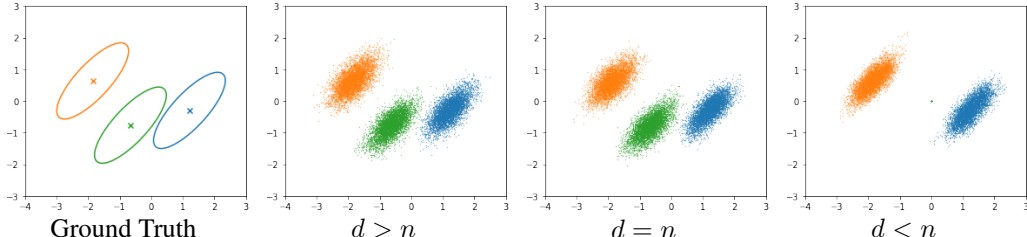

Figure 5: Analysis on amortization gap.

is trained with a standard stochastic optimization method like SGD. In GAN, the discriminator is also trained with standard stochastic optimization, which corresponds to the MLE case where noise injection is omitted. Although there are some investigations to apply Bayesian approach to GAN (Saatci & Wilson, 2017; He et al., 2018), their discriminators are not defined as energy functions.

**Wasserstein GANs** (WGANs) (Arjovsky et al., 2017) also have a loss function similar to LAE's:

$$\mathcal{L}_{\text{WGAN}}\left(\boldsymbol{\Theta}, \boldsymbol{\Psi}\right) = -\sum_{i=1}^{n} f_{\mathbf{x}}\left(\boldsymbol{x}^{(i)}; \boldsymbol{\Theta}\right) - f_{\mathbf{x}}\left(f_{\mathbf{x}|\mathbf{u}}\left(\boldsymbol{u}^{(i)}; \boldsymbol{\Psi}\right); \boldsymbol{\Theta}\right), \qquad (34)$$

where $D$ denotes the discriminator of WGANs that maps from the observation space into the real space $\mathbb{R}$. In this case, the counter part of the energy function is $-D\left(\boldsymbol{x}; \boldsymbol{\Theta}\right)$, although $D$ has a constraint of 1-Lipschitz continuity, which the energy function of LAE does not has.

## E   ADDITIONAL EXPERIMENT

We perform an additional experiment to investigate the amortization gap when the capacity of the inference model is not enough to meet the second condition of Theorem 1. We use the same experimental setting with the bivariate Gaussian example in Section 6.1, and change the dimensionality of the last linear layer of the inference model from 2 ($< d$) to 128 ($> d$). The results are summarized in Figure 5. It can be observed that the sample quality is good when the dimensionality of the last linear layer is equal to or greater than the number of data points (i.e., $d \geq n$). When the dimensionality is smaller than the number of data points, the samples for some data points shrink to a small area, while good samples are obtained for the remaining data points.

## F   EXPERIMENTAL SETTINGS

### F.1   CONJUGATE UNIVARIATE GAUSSIAN EXAMPLE

In the experiment of conjugate univariate Gaussian example, we initially generate 100 synthetic data $\boldsymbol{x}^{(1)}, \ldots, \boldsymbol{x}^{(100)}$, where each $\boldsymbol{x}^{(i)}$ is sampled from a univariate Gaussian distribution as follows:

$$p\left(z\right) = \mathcal{N}\left(z; \mu_{\mathbf{z}}, \sigma_{\mathbf{z}}^2\right), \quad p\left(x \mid z\right) = \mathcal{N}\left(x; z, \sigma_{\mathbf{x}}^2\right).$$

In this experiment, we set $\mu_{\mathbf{z}} = 0, \sigma_{\mathbf{z}}^2 = 1, \sigma_{\mathbf{x}}^2 = 0.01$. In this case, we can calculate the exact posterior as follows:

$$p\left(z \mid x\right)$$
$$= \mathcal{N}\left(z; \frac{1}{\frac{1}{\sigma_{\mathbf{z}}^2} + \frac{1}{\sigma_{\mathbf{x}}^2}}\left(\frac{\mu_{\mathbf{z}}}{\sigma_{\mathbf{z}}^2} + \frac{x}{\sigma_{\mathbf{x}}^2}\right), \left(\frac{1}{\sigma_{\mathbf{z}}^2} + \frac{1}{\sigma_{\mathbf{x}}^2}\right)^{-1}\right)$$

In this experiment, we obtain 20,000 samples using SGALD. We use four fully-connected layers of 128 units with tanh activation for the inference model and set the step size $\eta_\phi$ to 0.001. We set the batch size to 10.

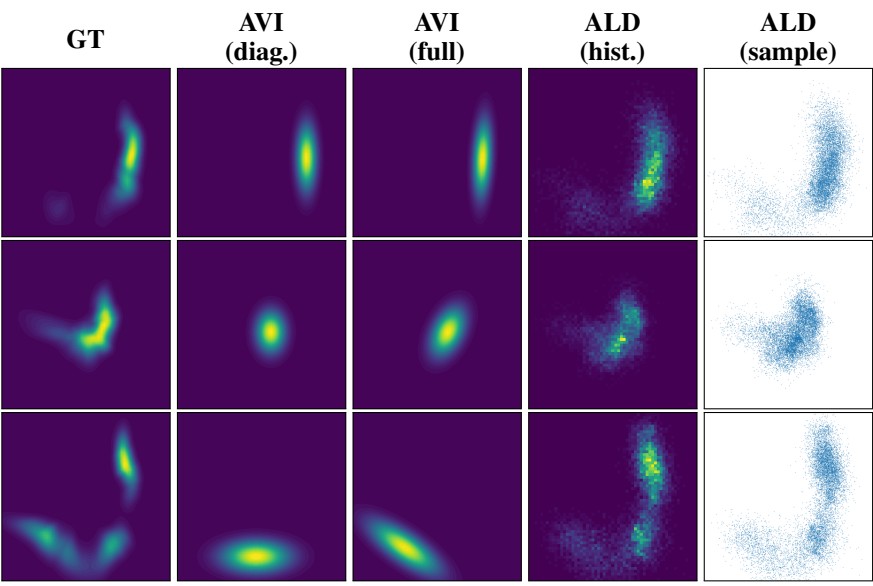

Figure 6: Neural likelihood experiments.

Table 2: Neural network architectures

| Encoder | Decoder |
|---|---|
| $\boldsymbol{x} \in \mathbb{R}^{d_{\mathbf{x}}}$ | $\boldsymbol{z} \in \mathbb{R}^{d_{\mathbf{z}}}$ |
| $\rightarrow \text{Conv3x1x2x64} \rightarrow \text{Conv4x2x2x128}$ | $\rightarrow \text{Deconv6x1x0x512}$ |
| $\rightarrow \text{Conv4x2x2x256} \rightarrow \text{Conv4x2x2x512}$ | $\rightarrow \text{DeConv4x2x0x256} \rightarrow \text{DeConv4x2x0x128}$ |
| $\rightarrow \text{Conv6x1x0x}n \rightarrow \text{Linear } d_{\mathbf{z}}$ | $\rightarrow \text{DeConv4x2x0x64} \rightarrow \text{Conv3x1x0x}d_{\mathbf{x}}$ |

| Energy | Sampler |
|---|---|
| $\boldsymbol{z} \in \mathbb{R}^{d_{\mathbf{z}}}$ | $\boldsymbol{u} \in \mathbb{R}^{d_{\mathbf{u}}}$ |
| $\rightarrow \text{Linear2048}$ | $\rightarrow \text{Linear2048} \rightarrow \text{Linear } n$ |
| $\rightarrow \text{Linear1}$ | $\rightarrow \text{Linear } d_{\mathbf{z}}$ |

## F.2 NEURAL LIKELIHOOD EXAMPLE

We perform an experiment with a complex posterior, wherein the likelihood is defined with a randomly initialized neural network $f_\theta$. Particularly, we parameterize $f_\theta$ by four fully-connected layers of 128 units with ReLU activation and two dimensional outputs like $p(\mathbf{x} \mid \boldsymbol{z}) = \mathcal{N}\left(f_\theta(\boldsymbol{z}), \sigma_x^2 I\right)$. We initialize the weight and bias parameters with $\mathcal{N}(0, 0.2I)$ and $\mathcal{N}(0, 0.1I)$, respectively. In addition, we set the observation variance $\sigma_x$ to 0.25. We used the same neural network architecture for the inference model $f_\phi$. Other settings are same as the previous conjugate Gaussian experiment.

The results are shown in Figure 6. The left three columns show the density visualizations of the ground truth or approximation posteriors of AVI methods; the right two columns show the visualizations of 2D histograms and samples obtained using ALD. For AVI method, we use two different models. One uses diagonal Gaussians, i.e., $\mathcal{N}\left(\mu(\boldsymbol{x};\boldsymbol{\phi}), \text{diag}\left(\sigma^2(\boldsymbol{x};\boldsymbol{\phi})\right)\right)$, for the variational distribution, and the oher uses Gaussians with full covariance $\mathcal{N}\left(\mu(\boldsymbol{x};\boldsymbol{\phi}), \Sigma(\boldsymbol{x};\boldsymbol{\phi})\right)$. From the density visualization of GT, the true posterior is multimodal and skewed; this leads to the failure of the Gaussian AVI methods notwithstanding considering covariance. In contrast, the samples of ALD accurately capture such a complex distribution, because ALD does not need to assume any tractable distributions for approximating the true posteriors. The samples of ALD capture well the multimodal and skewed posterior, while Gaussian AVI methods fail it even when considering covariance.

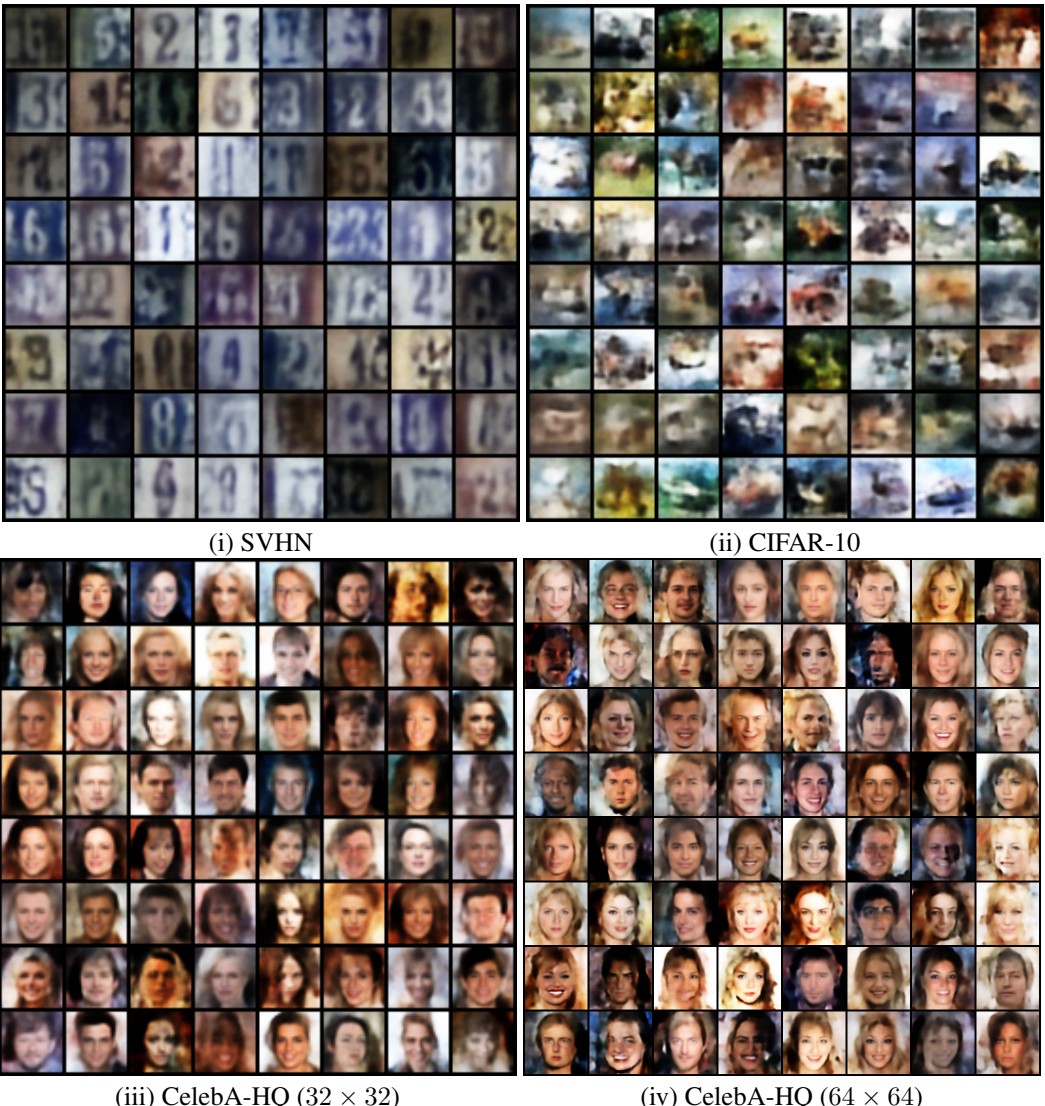

(i) SVHN          (ii) CIFAR-10

(iii) CelebA-HQ ($32 \times 32$)          (iv) CelebA-HQ ($64 \times 64$)

Figure 7: Generated images by LAE.

### F.3   IMAGE GENERATION

Our implementation of baseline models is based on the official public code of Pang et al. (2020a) which is available at `https://github.com/bpucla/latent-space-EBM-prior`. The architecture of neural networks for LAE is summarized in Table 2. Conv $k$ x $s$ x $p$ x $c$ denotes a convolutional layer with $k \times k$ kernel, $s \times s$ stride, $p \times p$ padding, and $c$ output channels. Deconv $k$ x $s$ x $p$ x $c$ denotes a transposed convolutional layer with $k \times k$ kernel, $s \times s$ stride, $p \times p$ padding, and $c$ output channels. Linear $d$ is a fully connected layer of output dimension $d$. We apply the swish function (Ramachandran et al., 2017) as activation after each convolution, transposed convolution or linear layer except the last one. $d_{\mathbf{x}}$, $d_{\mathbf{z}}$ and $d_{\mathbf{u}}$ are the dimensionality of $\mathbf{x}$, $\mathbf{z}$ and $\mathbf{u}$ respectively. We fix the parameters in the encoder and the sampler except for the last linear layer to meet the condition of Theorem 1. For all datasets, we set the minibatch size $m$ to 100. The latent dimensionality $d_{\mathbf{z}}$ is set to 64 for SVHN and 128 for CIFAR10 and CelebA-HQ. We set $d_{\mathbf{u}} = 2048$ throughout the experiments. The training of LAE is performed via the Langevin sampling iterations as explained in Section 4. We use preconditioned stochastic gradient Langevin dynamics (pSGLD) (Li et al., 2016) as the sampling algorithm. The implementation is available at `https://bit.ly/2Swow0F`

