# OpenReview forum: "Langevin Autoencoders for Learning Deep Latent Variable Models"
_ICLR.cc/2022/Conference — ICLR 2022 Submitted_

### Official Review · Reviewer_vRp6 · 2021-10-31

**Correctness:** 3
**Technical Novelty And Significance:** 3
**Empirical Novelty And Significance:** 3
**Recommendation:** 5
**Confidence:** 3

**Main Review:**

I enjoyed the overall paper. I felt the presentation was perhaps a little unecessarily complex. I have a few technical concerns

# Good points
- I felt the writing and presentation of existing literature was generally very good, clear and concise.

- the theoretical concept "one chain for all datapoints" (and linear projections) also feels like a good idea to me. The theorem requires that all samples for all posteriors $z^{(i)}$ must be varying linear projections ($g(x)$) of a single communal chain $\Phi$. Or alternatively, all data item specific posterior distributions are linear projections of a single higher dimensional distribution. This sounds really cool!

- I enjoyed the theorem and the discussion of the rank of the G matrix.


# Concerns

- In algorithm 2, the generative model weights, $\Theta$, are also sampled using the Euler-Maruyama method, however this does not correspond to the theorem 2.1 which requires that the generative model $\Theta$ is fixed (otherwise there is no true posterior to approximate). (Upon further inspection, it looks like the given loss function (14) is the evidence lower bound of the data and hence sampling model weights is similar to Bayesian neural network sampling of weights with the ELBO and not model evidence).

- P4, line 3: Test time sampling: why not pregenerate a markov chain during/after training that can be reused again and again at test time? Surely the whole point of amortization is the “one-chain fits-all” hypothesis? If the authors propose resampling at test time then the proposed method is not strictly amortized and there is no actual speedup?

- In the experiments, would it be possible to also report estimated marginal dataset likelihood $\log p(x)$? There are many generative modelling works that use this as a metric (e.g. PixelCNN).

- I am surprised by the *extremely* high variance of of the Frechet Inception distance among baselines, I struggle to believe that all the standard methods are *that* bad, presumably either the error bars are not accurate or the implementation of baselines is not ideal. For me, unfortunately, this make the paper hard to accept. (I realise this may contradict other venues/reviewers that emphasise empirical results).

- in the experiments, I think it is valuable to know why LD and LD+EMB are worse than LAE? Surely LD + EMB is the gold standard (one MCMC chain per dataitem) that the LAE strives to achieve, while this may be computationally expensive, I believe that such a baseline should at least be implemented to outperform (or match) LAE, (i.e. agreeing with the theory) this could also show the MCMC amortization gap that this new amortization method introduces (again agreeing with the theory). Currently these are unanswered question that feels rather central to the contribution of the paper.

- regarding Theorem 2.1, and the implication that all posteriors are linear projections of a bigger common distribution, this feels very sensitive to the big distribution dimension, currently this is not properly investigated. If the dimension is large, then the communal MCMC chain is no cheaper than parallelized low dim chains across dataitems, if the dimension is too small, this imposes too much similarity across all the posteriors and the method suffers, this trade off is not clear.

# Minor Points

- there appears almost no mention of Bayesian neural networks. MCMC sampling of network weights has been extensively studied, the proposed method is also MCMC sampling network weights while just using a different  distribution to sample from. Presumably, any Bayesian neural network approach for the encoder may be used?

- If I understand correctly, training is using the Eurler Maruyama method and so the algorithm simplifies to SGD with Gaussian noise, presumably this makes convergence much slower than a VAE or VAE-flow which may use RMSprop or Adam or any method with momentum across minibatches?

- (very minor) regarding the disadvantages of VAEs, the ELBO training objective is the KL divergence of true and approximate posterior, i.e. it makes a inference network Gaussian that looks like the posterior but it also, *theoretically*, it makes a posterior that looks Gaussian,  hence the claims that VAEs are existentially flawed feel slightly overstated to me, though I realise this is not a common opinion and in practice this is indeed very often not the case.

- the presentation of the proposed method seems a little overly complex or difficult to follow, as I have described, I personally found it much easier to think in terms of a communal chain and linear projections, I am sure there must be many other intepretations that are simpler and easier to follow than the paper in its current form.


I enjoyed the paper and I am less well read in the area hence absolute novelty I cannot confidently judge (I wouldn't be surprised if this idea is well established in the more pure "non-deep" statistics community) though I do believe the idea has merit and is worth accepting. However the rather chaotic numerical results are very disturbing.


**Summary Of The Paper:**

Typically, approximating the (analytically intractable) posterior of deep latent variable models can be done in two ways, variational inference methods learn an (analytically tractable) distribution to approximate the true posterior. For large datasets where each dataitem has its own posterior distribution, the parameters of the optimal approximate distribution may be predicted as a function of the dataitem, the comon practice of “amortization”. Thus the approximate posterior is quick to infer at test time however it is also limited by the flexibility of the familty of distributino used for the analytic approximation.

Meanwhile, Markov Chain Monte-Carlo methods generate a collection of samples to approximate true posterior. As each dataitem in a large dataset has a unique posterior, each would require its own Markov chain of samples which is a more accurate approximation, however is time consuming.

The author propose a mixture of these approaches. At a very high level, the authors propose to combine a single “communal” Markov chain, $\Phi$, together with a function that maps dataitems into a weight vector, $g(x)$, that linearly maps the communal markov chain onto a markov chain that approximates the posterior for the given dataitem p(z|x)~$\Phi g(x)$. Replace one chain per item (traditional MCMC) with a communal chain and one linear mapping per dataitem, where the mapping is amortized.

The authors describe application of this new trick to latent space of an autoencoder latent variable model and the sampling from an unconditioned density such as an energy based model. The MCMC algorithm of choice is the well known Langevin Dynamics solved by the Euler-Maruyama method. The authors describe training in minibatches simulataneously regenerating the MCMC chains which reduces to standard SGD with noise. Toy experiments show proof of concept and image generation tasks show positive results as measured by Frechet Inception distance.


**Summary Of The Review:**

I like the paper, I feel that the writing could be simplified somewhat, the method treats latent posteriors over data points as unique linear projections of a global distribution over network final layer weights.

- to me, the proposed method is an instance Bayesian neural networks with a different distribution (the ELBO) over weights (eg https://arxiv.org/abs/1902.02476, http://proceedings.mlr.press/v48/gal16.pdf)

- the experimental results have very high variance amongst baselines, this is concerning, particularly, the "truth" that the proposed method aims to approximate performs *worse* that the approximation.

- treating all dataitem posteriors as a linear projections of a single communal distribution seems risky, further comment would be appreciated.

---

> ### Author Response · Authors · 2021-11-18
> **Response to Reviewer vRp6 (1/2)**
>
> Thank you for your detailed comments.
> We answer your comment to address your concern.
> We also have some updates on our paper, so please check it too.
>
> **Is the generative model $\Theta$ a Bayesian neural network in Section 4?**
>
> Yes. In Section 4, we took a Bayesian approach, where the learning algorithm is formulated as the calculation of the joint posterior distribution $p \left( \Theta, Z \mid X \right)$; hence the generative model $\Theta$ is defined as a Bayesian neural network.
> Why we took the Bayesian formulation is that the learning algorithm can be more simply described than formulating it as maximization of the marginal likelihood.
> In the updated version, we have added the maximum likelihood version of LAE in Appendix B.
> In this version, the generative model $\Theta$ is fixed when performing ALD to get samples from the posterior.
>
> **Is it okay for the generative model $\Theta$ not to be fixed when performing ALD**
>
> As you mentioned, Theorem 1 assumes that the generative model $\Theta$ is fixed when performing ALD to obtain samples from posterior.
> However, even when we also update $\Theta$ using LD in addition to $\Phi$, the samples of $\Theta$ and $\Phi$ converge to the target distribution, although the target here is the joint posterior $p \left( \Theta, Z \mid X \right)$ instead of $p \left( Z \mid X, \Theta \right)$.
> In the proof of Theorem 1, we first consider the stationary distribution of $\Phi$ and transform it into the stationary distribution of $Z$ by the change of random variable (or the pushing forward measure).
> We can take the same strategy to prove the convergence of $(\Theta, Z)$ into the joint posterior $p \left( \Theta, Z \mid X \right)$ by considering the stationary distribution of $(\Theta, \Phi)$ and transforming it into $(\Theta, Z)$ by their pushing forward measure.
>
> **ALD in test time**
>
> In general, we cannot access test data during training, so we have to run additional MCMC to obtain samples of the posterior for new test data $p \left( z \mid x_{\mathrm{test}} \right)$.
> In a naive way, we randomly choose an initial sample and run a Markov chain iteratively, but it takes a long time to converge.
> We can fasten the convergence by initializing the sample using the inference model that has been used for posterior sampling during training.
> Therefore, ALD is also useful in test time to fasten the convergence of MCMC, which amortizes the cost of running a naive MCMC.
> To clarify it, we also provide the algorithm of ALD in test time in Algorithm 3 in the appendix.
>
> **Evaluation of marginal likelihood**
>
> We agree that it is desirable if we can compare the performance using the marginal likelihood.
> Unfortunately, it is difficult to compute the marginal likelihood (or its lower bound) for the LAE, because the inference model is a deterministic function, which cannot be used to calculate the ELBO.
> We also tried to use some techniques to calculate the marginal likelihood (e.g., annealed importance sampling), but its variance was too big to be used for the comparison with other baseline models, so we gave it up.
> Developing a comprehensive framework to evaluate many types of deep generative models is also an important direction of future research.
>
> **On empirical results**
>
> Based on your feedback, we have reimplemented the baseline models using the official code of Pang et al. (2020a), which is available at https://github.com/bpucla/latent-space-EBM-prior.
> We found that the high variance of the baseline results was due to the too big step size of LD sampling, which deteriorated the sample quality of non-amortized LD methods.
> By this update, the results of baseline models have been greatly improved, and we find that LEBM outperforms our LAE in SVHN and CelebA-HQ.
> As you mentioned, it is natural that LEBM could perform better than our LAE, because LEBM conducts exact sampling from the posterior and the EBM, whereas LAE's amortized sampling may have some approximation error.
> Hence, the advantage of LAE over LEBM is still the efficiency of sampling by the power of amortization.
> The reason LAE outperforms ABP (LD without amortization) is that ABP does not use EBM for the latent prior, which makes its expressive power limited compared to LAE (and LEBM).
>
> **Effect of amortization gap**
>
> As you pointed out, it may be difficult to make the dimension of the linear projection sufficiently large especially when the data size is huge, so investigating what will happen when the dimension is too small is important.
> To empirically analyze it, we have added an experiment on 2d toy examples, changing the dimension from small to large values.
> The result is shown in Appendix E.
> We observe that, when the dimension is too small, samples are still good for some data points, but ones for the remaining data points collapse to a certain area.

---

> > ### Author Response · Authors · 2021-11-18
> > **Response to Reviewer vRp6 (2/2)**
> >
> > **Is there the RMSProp (or Adam) version of Euler–Maruyama method?**
> >
> > As we mentioned in the last paragraph of Section 3.1, we use preconditioned SGLD (pSGLD) rather than naive SGLD (Euler–Maruyama method) to improve the convergence of stochastic MCMC.
> > The relationship between naive SGLD and pSGLD is almost identical to SGD and RMSProp; pSGLD scales the step size considering the curvature of the parameter space instead of using a fixed step size.
> >
> > **Disadvantage of VAE**
> >
> > As you pointed out, the training objective of VAE is to maximize $\log p \left( x ; \Theta \right) - D_{\mathrm{KL}} \left( q \left( z \mid x \right) \|| \ p \left( z \mid x ; \Theta \right) \right)$.
> > This means that VAE's posterior $p \left( z \mid x ; \Theta \right)$ is constrained to be close to Gaussian (if you choose it as the variational distribution $q$).
> > This is a strong constraint because the true posterior will be Gaussian if and only if the decoder is a linear function.
> > Hence, this regularization makes the VAE as close to the linear model as possible (of course, it does not actually become linear, because it also maximizes the log marginal likelihood in addition to the KL constraint).
> > When the variational distribution $q$ has more expressive power (e.g., using normalizing flows), the effect of this constraint will be small.
> >
> > **Other interpretation of ALD**
> >
> > We agree that ALD could be interpreted in other ways and analyzing the algorithm from other perspectives will be interesting future work.
> > The reason we explain ALD in a current manner is that the connection to amortized variational inference (AVI) can be clearly described (i.e., replacing the optimization/sampling of local parameters into that of global parameters shared across all data points).
> >
> > We would be glad to respond to any further questions and comments that you may have.
> >
> > Thanks.

---

> > > ### Comment · Reviewer_vRp6 · 2021-11-22
> > > **Thank you for the thoughtful replies!**
> > >
> > > - **RMSprop** thank you for the description of preconditioning in SGD, this is not something I am familiar with yet if it has already been somewhat taken into account, then I retract my initial comment.
> > >
> > > - **chaotic baselines** thank you for the updated table of results, I find it reassuring to see the baseline performing better in 3 of the 4 experiments and significantly increases my faith in the results. A quick glance at the changes to the paper it seems that these negative (and valuable) results are not discussed?
> > >
> > > - **Disadvantage of VAE** thank you for the insightful comment, indeed a linear decoder would result in true Gaussian posterior (assuming Gaussian likelihood)
> > >
> > > - **ALD at Test time** may be I misunderstood, I believe the amortization is a result of using a "Bayesian" final layer of the encoder: MCMC sample lots of final layer encoder weights $\Phi$, these final layer weights come from an underlying distribution that is *marginalized* over data points $\sum_i (z_i, x_i)$, these MCMC samples are supposedly *not* dataitem specific, rather it is the linear mapping $g(x)$ that is data specific. Hence, one may use the training set to sample many many $\Phi$ and use them again at test time. (The Langevin implementation conflates minibatches with samples and momentum would reduce this)
> > >
> > > - **Amortization Gap experiments** Thank you for the added results on the toy datasets. This does seem rather intuitive, running one chain per dataitem and concatenating them is theoretically equivalent to a single  $n * dim(z)$ dimensional chain. In the dim(z)=2 case with $n=3$, I would expect $d=6$ to be the largest necessary $\Phi$ chain dimension and performance to decay as d decreases, which is shown in the experiment. I feel this experiment and the authors comments upon $d$ should be included in the main paper (I didn't see it in the red text anyway), a very low $d$ can significantly undermine the method and how this works in larger experiments is still an open (and also not easy) question.
> > >
> > > I have increased my score to reflect the changes.

---

> > > > ### Author Response · Authors · 2021-11-24
> > > > **Reply to Comments by Reviewer vRp6**
> > > >
> > > > Thank you for your further comments!
> > > >
> > > > **chaotic baselines**
> > > >
> > > > > A quick glance at the changes to the paper it seems that these negative (and valuable) results are not discussed?
> > > >
> > > > About the negative results, we have added the statement "It can be observed that LAE outperforms VI-based methods in terms of FID, although it does not reach LEBM’s performance in SVHN and CelebA-HQ." in the last paragraph of Section 6.2 in the updated version (sorry that we forgot to change the text color into red).
> > > >
> > > > **ALD at test time**
> > > >
> > > > Your understanding is mostly correct. The main point of ALD is to represent the sampling process of the latent posterior by running the update iterations of the encoder weights $\Phi$ and collecting the outputs of the encoder for each data point.
> > > >
> > > > > Hence, one may use the training set to sample many many $\Phi$ and use them again at test time.
> > > >
> > > > It is possible to perform test time sampling in that way if we can access test data during training or can store all $\Phi$ at each iteration, but it is difficult in practice, we think.
> > > > Alternatively, we have the encoder, whose weights are fixed after the training, and we can use it to initialize the MCMC for the test data as we described in Algorithm 3 in the appendix.
> > > > Another possible way of test time sampling is to update the encoder weights again using test data after the training.
> > > > This may be better than just using the encoder for initialization as in Algorithm 3 especially when we have a lot of test data.
> > > >
> > > > **Amortization Gap experiments**
> > > >
> > > > $d$ is the dimensionality of $g \left( x \right)$ as described in Theorem 1, so the dimension of $\Phi$ will be $\mathrm{dim} \left( z \right) \times d$.
> > > > Hence, $d = 3 (= n)$ is the largest necessary dimension to meet the second condition in Theorem 1.
> > > > We will also add the definition of $d$ in Appendix E in the final version for clarification.
> > > >
> > > > Feel free to ask more questions if you have.
> > > >
> > > > Thanks.

---

> > > > > ### Comment · Reviewer_vRp6 · 2021-11-28
> > > > > **Thank you for the clarity, however......**
> > > > >
> > > > > **ALD at test time**
> > > > > Sorry for repeating this but I would like to really pin down my understanding.
> > > > >
> > > > > I am only considering the case without minibatching, the case where $batchsize=n$ as implied by the equations in the paper, then MCMC iterations would be sampling weights from a single deterministic posterior density over $\Phi$.
> > > > >
> > > > > In standard training of any neural network, the loss is an expectation over some unobserved oracle $P(x)$ that is instead approximated by MC with actual training samples $x^{(1)},.....,x^{(n)}$. It is also typically assumed that the test set is also sampled from the same oracle $P(x)$ (assuming otherwise would be the field of outlier detection/domain shift etc).
> > > > >
> > > > > The only difference in posterior over $\Phi$ is from swapping $x^{train}$ to $x^{test}$, which are both finite sample approximations of a common $P(x)$. If $n$ becomes infinitely large, the train and test posteriors over $\Phi$ would be identical, there is no need to redo MCMC at test time with test data.
> > > > >
> > > > > Hence, performing MCMC over weights at test time is to account for the finite sample error between train set and test set, i.e. to "overfit" to the "noise" in the realization of the test set.
> > > > >
> > > > > I feel that you seem to believe test data is fundamentally required at test time. I am still struggling to see why. In the traditional case of one MCMC per dataitem, indeed individual dataitems are required, but when using MCMC over network weights, the posterior over $\Phi$ is defined by a marginalization over the oracle $P(x)$, hence test data is not _required_ and to use test data for further MCMC does have a feeling of overfitting, learning the test set specific noise. Would the you be able to confirm this?
> > > > >
> > > > >
> > > > > **Amortization Gap experiments**
> > > > > Ah, sorry for my confusion about $d$, indeed I was referring to the dimension of $\Phi$, and $d=n$ is indeed what I understand, i.e. the equivalent dimension of doing one chain per datapoint. And the condition of Theorem 1 makes perfect sense, any amortized MCMC chain over network parameters  **must** have the same dimension as doing one MCMC chain per datapoint, i.e. no dimensions are lost going from "amortized" MCMC to full MCMC and back, if I have one more dataitem, I need more network weights to accommodate one more latent MCMC chain.
> > > > >
> > > > > Hence these above two comments together seem to point towards a bit of design contradiction, learning network weights requires an average over a dataset, weights are *not* dataitem specific as shown by the graphical model in Fig 1(a), **$d$ cannot depend upon $n$**. The theorem requirement 2 states that more data requires more network weights, the encoder architecture must grow with datapoints and hence weights are dataitem specific, **$d\geq n$**.
> > > > >
> > > > > If this interpretation is accurate, then I feel the theoretical contribution is somewhat underwhelming (i.e. it's a "no brainer") and the contribution of the paper is largely an empirical demonstration of how a $d<n$ still works which again isn't too surprising given that similar data items may have similar posteriors and a full set of $n$ MCMC chains has redundancy and can be replaced by one chain of $d<n$, I feel exploiting this fact is certainly a valuable contribution and worth pursuing! Unfortunately I still feel the presentation of the paper could be made much clearer (again assuming I haven't misunderstood!) the paper would need significant rewriting
> > > > >
> > > > > Do you agree with my assessment of the seemingly opposing requirements for $d$?
> > > > >
> > > > > I have left my score unchanged.

---

> > > > > > ### Author Response · Authors · 2021-11-29
> > > > > > **Reply to Feedback by Reviewer vRp6**
> > > > > >
> > > > > > Thank you for your comments!
> > > > > >
> > > > > > **ALD at test time**
> > > > > >
> > > > > > > The only difference in posterior over $\Phi$ is from swapping $x^{train}$ to $x^{test}$, which are both finite sample approximations of a common $P(x)$. If $n$ becomes infinitely large, the train and test posteriors over $\Phi$ would be identical, there is no need to redo MCMC at test time with test data.
> > > > > >
> > > > > > > Hence, performing MCMC over weights at test time is to account for the finite sample error between train set and test set, i.e. to "overfit" to the "noise" in the realization of the test set.
> > > > > >
> > > > > > Yes, that is exactly what we meant. If we can expect that the samples of $\Phi$ that are obtained using training data can also generalize to test posterior, it is okay to use them as test time sampling. So, the algorithm we described in Algorithm 3 is a conservative approach that does not expect such generalization.
> > > > > >
> > > > > > > when using MCMC over network weights, the posterior over $\Phi$ is defined by a marginalization over the oracle $P(x)$
> > > > > >
> > > > > > We think that is not correct.
> > > > > > The posterior over $\Phi$ is defined for finite training samples; otherwise, the matrix $G$ in Theorem 1 cannot be well-defined.
> > > > > > What Theorem 1 guaranttees is the convergence of ALD samples for *training posteiors* (not all posteriors over the oracle $P(x)$).
> > > > > > Hence, it may cause overfitting to training posteriors.
> > > > > > The same discussion is applicable to amortized variational inference (AVI).
> > > > > > In AVI, the encoder is trained to minimize KL divergence between variational posteriors and true posteriors *for training data*. Therefore, the encoder's inference for test data may be suffered from overfitting (e.g., see [1]).
> > > > > >
> > > > > > **Amortization Gap experiments**
> > > > > >
> > > > > > Yes, we agree all comments on the amortization gap.
> > > > > >
> > > > > > > Hence these above two comments together seem to point towards a bit of design contradiction, learning network weights requires an average over a dataset, weights are not dataitem specific as shown by the graphical model in Fig 1(a), $d$ cannot depend upon $n$.
> > > > > >
> > > > > > It seems there is a slight misunderstanding.
> > > > > > It is true that the weights of the generative model (decoder) cannot depend upon the data size $n$, because the generative model defined by the graphical model in Fig 1(a) is fixed regardless of the data size $n$.
> > > > > > However, the weights of the inference model (encoder) are not restricted by the graphical model; they can be chosen depending on the data size.
> > > > > > Note that the encoder is not included in the graphical model, because it does not affect the data generating process assumed by the graphical model.
> > > > > >
> > > > > > > The theorem requirement 2 states that more data requires more network weights, the encoder architecture must grow with datapoints and hence weights are dataitem specific, $d \geq n$.
> > > > > >
> > > > > > That is correct. So we have a trade-off between the sample quality and the computational costs of the encoder as described in Section 3.2.
> > > > > >
> > > > > > > If this interpretation is accurate, then I feel the theoretical contribution is somewhat underwhelming (i.e. it's a "no brainer")
> > > > > >
> > > > > > We agree that the result itself is not so surprising (it exactly matches the intuition).
> > > > > > But we believe the theoretical contribution is not so minor because its derivation is not necessarily easy.
> > > > > > Moreover, Theorem 1 is essential to show that ALD is a valid MCMC algorithm; otherwise ALD loses theoretical basis to be used for the training of latent variable models.
> > > > > >
> > > > > > > Do you agree with my assessment of the seemingly opposing requirements for $d$?
> > > > > >
> > > > > > As we mentioned above, we agree that the second condition in Theorem 1 requires the weights of the inference model to have a large dimensionality, but it does not contradict with the graphical model in Fig 1(a), because the encoder's dimension can be chosen without depending on the graphical model.
> > > > > >
> > > > > > If you still have any concerns, please ask again.
> > > > > >
> > > > > > Thanks!
> > > > > >
> > > > > > **Reference**
> > > > > >
> > > > > > [1] Shu, Rui, et al. "Amortized inference regularization." Proceedings of the 32nd International Conference on Neural Information Processing Systems. 2018.

---

> > > > > > > ### Comment · Reviewer_vRp6 · 2021-11-29
> > > > > > > **Further Clarification**
> > > > > > >
> > > > > > > I misunderstood the model, and thank you for the clarification.
> > > > > > >
> > > > > > > However, requirement 2 of the theorem implies states that the proposed amortized MCMC must have equal or higher dimension than many individual MCMC chains put together. So what  is being amortized? Where is the computational saving?
> > > > > > >
> > > > > > > In Section 3.2 it states
> > > > > > >
> > > > > > > "experiments, we confirm that preserving the condition not become a significant computational overhead in practice."
> > > > > > >
> > > > > > > So if we have the same MCMC dimensions and we are not assuming generalizability, why not do one MCMC chain per dataitem?
> > > > > > >
> > > > > > > Hypothetically, if $g(x)$ were learnable, then it could just learn to map each of $n$ dataitems into a $n$ dimensional one-hot vector, and simply cherry pick one row out of all of the $\Phi$ matrix. If there is a MCMC chain of $\Phi$, each row is a single MCMC chain of $dim(z)$ bespoke to a single dataitem, i.e. it reduces to standard MCMC.
> > > > > > >
> > > > > > > Again, if the train-> test generalization assumption is not being used, and MCMC dimension is not reduced (or rather increased), then what is the point of amortization?
> > > > > > >
> > > > > > > Thank you!

---

> > > > > > > > ### Author Response · Authors · 2021-11-29
> > > > > > > > **Reply to Reviewer vRp6**
> > > > > > > >
> > > > > > > > Thank you for your quick reply.
> > > > > > > >
> > > > > > > > **What is the point of amortization?**
> > > > > > > >
> > > > > > > > As we described in the paragraph starting from "By this amortization, ..." in Section 3.1, the main advantage of amortization is that it can leverage minibatch training, which reduces the cost of evaluating the potential energy $U (x, z)$ for all data points by substituting it with its minibatch statistics.
> > > > > > > > This characteristic is important to allow MCMC-based DLVM training to scale to large-scale datasets.
> > > > > > > >
> > > > > > > > Of course, when we can expect the "train-> test generalization", it is also an advantage of ALD. So we have provided Algorithm 3 as an example of a test time algorithm, which uses the trained inference model to initialize standard MCMC to accelerate the convergence.
> > > > > > > >
> > > > > > > > Again, the same discussion is applicable to AVI (and VAE).
> > > > > > > > AVI leverages the inference model to accelerate the optimization of variational parameters via minibatch training, and the trained inference model can be also used for the test time inference (expecting generalization).
> > > > > > > > In addition, what makes VAE so popular is, we think, its simplicity: a stochastic autoencoder that is trained via the maximization of a single ELBO objective yields a generative model.
> > > > > > > > We believe that our LAE also has such simplicity: a deterministic autoencoder trained with SGLD yields a generative model (though it requires some assumptions as described in Theorem 1), which potentially makes MCMC-based DLVM training more tractable for practitioners.
> > > > > > > > We have described this motivation in the third paragraph of the introduction that starts from "As in VI, ..." in the paper.
> > > > > > > >
> > > > > > > > Feel free to ask more questions if you have.
> > > > > > > >
> > > > > > > > Thanks!

---

> > > > > > > > > ### Comment · Reviewer_vRp6 · 2021-11-29
> > > > > > > > > **Questionable Purpose of Amortization**
> > > > > > > > >
> > > > > > > > > In the paper you say
> > > > > > > > > "A significant advantage of amortization is that the cost of MCMC can be reduced by using minibatch training"
> > > > > > > > >
> > > > > > > > > After this discussion I am less convinced that this is true.
> > > > > > > > >
> > > > > > > > > The Theorem requires that the output of $g(x)$ produces linearly independent vectors of dimension $d\geq n$ for each data item $x$, the matrix $G$ is rank $n$. Let $d=n$, this means that I can linearly transform $G$ into the $n*n$ identity matrix and back with no information lost. So I may simply incorporate this as an extra linear layer (with no activation function) in $g(x)$ so that it outputs unique $n$ dimensional one-hot encodings (as $g(x)$ is randomly initialized we are free to change it as we please). In this case, the matrix multiplication $\Phi g(x_i)$ simply takes the $n$ columns of $\Phi$ and cherry picks out the $i$th column to be a latent sample point $z_i$. The proposed algorithm then simply performs MCMC on the $i$th column, $z_i$, everytime it sees the $x_i$ dataitem, and otherwise the $i$th column is never updated (it is not used in $U()$ and thus has no gradient, it is only updated by the  uninformative noise). Thus (minibatch or not) MCMC over $\Phi$ simply reduces to each column becoming its own MCMC chain $z_i$ for dataitem $x_i$ (albeit with the uninformative Gaussian noise added in).
> > > > > > > > >
> > > > > > > > > In other words, I believe MCMC over $\Phi$ with $n$ rank $G$ are the same as MCMC per dataitem for training (it is just doing MCMC per dataitem in a rotated latent space). If $d>n$, it is more expensive than MCMC per dataitem.
> > > > > > > > >
> > > > > > > > > Again, this is not a surprise and is perfectly inline with the theorem which I interpret as "what do I need to do to make sure the ALD is not worse than MCMC per dataitem?" and the answer is "use just as much computation as doing MCMC per dataitem".
> > > > > > > > >
> > > > > > > > > In a normal VAE, the encoder output may be used as a warm start to finding optimal variational parameters for further gradient ascent. The benefit of amortization is to get a near optimal posterior in "one-shot" and avoid much of the computation.
> > > > > > > > >
> > > > > > > > > In algorithm 3, the encoder is used to get an initial $z_j$ at test time from a test sample $x_j$. If the goal is to have a warm start for a making a new MCMC chain from scratch at test time, then the mean from a standard VAE encoder may be used within algorithm 3? Or any method to predict $z_j^{init}$ from $x_j$?
> > > > > > > > >
> > > > > > > > > Hence training would appear to be computationally comparable to MCMC per dataitem, testing by algorithm 3 reduces MCMC cost by removeing the "burn in" part of the MCMC chain but the rest of the chain is still computed. Hence I am still left wandering what the purpose of amortization is?

---

> > > > > > > > > > ### Author Response · Authors · 2021-11-30
> > > > > > > > > > **Answer to Concern about Purpose of Amortization**
> > > > > > > > > >
> > > > > > > > > > Thank you for your further comment.
> > > > > > > > > >
> > > > > > > > > > As you mentioned, ALD will be identical to datapoint-wise MCMC when $G$ is the $n \times n$ identity matrix; and in that case, ALD does not have any advantages by minibatch training because only partial elements of $\Phi$ are meaningfully updated.
> > > > > > > > > > In other cases, where G is a non-identity matrix, ALD may have the benefit of amortization via minibatch training.
> > > > > > > > > > Hence, the design of the feature extractor $g(x)$ is important to fasten the convergence by amortization; and what kind of structure is optimal for $g(x)$ is still an open problem, while it is clear that the case where $G$ is an identity matrix is the worst case.
> > > > > > > > > > We only have an empirical study regarding it in the paper, so theoretical discussion is a possible future work.
> > > > > > > > > > We have mentioned it in the last line of the conclusion in the updated paper.
> > > > > > > > > >
> > > > > > > > > > > In a normal VAE, the encoder output may be used as a warm start to finding optimal variational parameters for further gradient ascent. The benefit of amortization is to get a near optimal posterior in "one-shot" and avoid much of the computation.
> > > > > > > > > >
> > > > > > > > > > We agree, and the same can be said for ALD.
> > > > > > > > > > Because we have a trade-off between the sample quality and the capacity of the inference model as we stated in 3.2, we can obtain *near-optimal* posterior samples by choosing an appropriate size of an inference model, considering the computational cost.
> > > > > > > > > >
> > > > > > > > > > > In algorithm 3, the encoder is used to get an initial $z_j$ at test time from a test sample $x_j$. If the goal is to have a warm start for a making a new MCMC chain from scratch at test time, then the mean from a standard VAE encoder may be used within algorithm 3? Or any method to predict $z_j^{init}$ from $x_j$?
> > > > > > > > > >
> > > > > > > > > > Yes, and that is exactly what DLGM, which we used as a baseline in the experiment, is doing.
> > > > > > > > > > DLGM uses VAE's encoder to initialize posterior MCMC.
> > > > > > > > > >
> > > > > > > > > > Thanks.

---

> > > > > > > > > > > ### Comment · Reviewer_vRp6 · 2021-11-30
> > > > > > > > > > > **My intuition is that $G$ doesn't matter**
> > > > > > > > > > >
> > > > > > > > > > > My intuition is that the structure of $G$ makes no difference, as long as it is full rank.
> > > > > > > > > > >
> > > > > > > > > > > If $G$ is the identity matrix, it is clear that the $i$th column of $\Phi$ is a latent point $z_i$ for the sample $x_i$.
> > > > > > > > > > >
> > > > > > > > > > > We may perform one big MCMC over $\Phi^{n\times dim(z)}$, or we may perform one big MCMC over $Z = [z_1,...,z_n]\in \mathbb{R}^{n\times dim(z)}$, either can be done in minibatches or not. If G is the identity matrix, then both spaces are axis aligned, individual columns are updated in each batch and non-batch columns are left constant. If $G$ is not identity, then spaces are not axis aligned and MCMC over one $z_i$ (column of $Z$) corresponds to MCMC over some linearly independent subspace of $\Phi$ and "orthogonal" subspaces of $\Phi$ are left constant. My intuition is that they are the same MCMC and only differ by a rotation.
> > > > > > > > > > >
> > > > > > > > > > > I truly apologize if I am mistaken, and I hope future work can clarify this, but I still do not see how minibatching, or amortization makes a difference in the proposed method. Again, this makes perfect sense given the theorem, which I interpret as "there is no free lunch", G must be linear and full rank _purely because it is just a rotation_ of the original expensive MCMC problem.

---

> > > > > > > > > > > > ### Author Response · Authors · 2021-11-30
> > > > > > > > > > > > **You are correct, but...**
> > > > > > > > > > > >
> > > > > > > > > > > > Thanks for your reply.
> > > > > > > > > > > >
> > > > > > > > > > > > We understand your point, and we think you are correct: only orthogonal subspaces of $\Phi$ corresponding to minibatch are updated even when $G$ is not an identity matrix.
> > > > > > > > > > > > However, what is important here is that posterior samples of latent variables ($Z$) for *all data points* (not only minibatch) are updated when we update $\Phi$ using minibatch, and the samples surely converge to the true posteriors.
> > > > > > > > > > > > If we update $Z$ with datapoint-wise MCMC in a minibatch manner, only the minibatch part of $Z$ is updated; and the same happens if $G$ is the identity matrix.
> > > > > > > > > > > > This is why the structure of $g(x)$ is important: if we choose an appropriate form of $g(x)$, we can efficiently update the posterior samples for all data points only with minibatch statistics by making use of the correlation between data points through $g(x)$.
> > > > > > > > > > > >
> > > > > > > > > > > > Thanks!

---

> ### Author Response · Authors · 2021-11-27
> **A Gentle Reminder to Reviewer vRp6**
>
> Thank you again for your efforts in reviewing our paper and your constructive comments. We also appreciate your joining the active discussion here to make our paper better. The discussion period will end soon, so please let us know if you have further comments about our reply to your feedback.
>
> Thanks!

---

### Official Review · Reviewer_Lr3z · 2021-10-31

**Correctness:** 2
**Technical Novelty And Significance:** 2
**Empirical Novelty And Significance:** 2
**Recommendation:** 5
**Confidence:** 4

**Main Review:**

(+) The paper considers an important and challenging problem. The idea discussed is quite interesting and somewhat new.

However, there are some downsides of the current submission:

(-) The presentation of the paper needs to be improved. Quite a few places are unclear which need to be better motivated or to be further described. For example, what does $u$ represents in Eqn. 14? Any motivations/insights on why consider Langevin process on the parameter space (the parameter space usually has much higher dimensions compared to the latent codes itself)? How to choose $g(x)$ as in theorem 1? If only the last layer is trainable, does the inference model parametrized in this way has very limited expressive power? Also, in algorithm 2, all the update is on the parameter space including generator as well as encoder. The paper should also need to be well motivated on why performing Langevin on their parameter spaces (as in Eqn 10, 11) as well.

(-) The experiments need to be improved. The sample quality as well as score is relatively weak. Why not use reported FID of the baselines in table 1? For example, the LEBM achieves lower FID on cifar10 on their paper, why not choose their model structure and retrain the proposed model for the comparison?

**Summary Of The Paper:**

The paper proposes to use amortized MCMC to learn latent space energy-based model (EBM). Specifically, instead of running Langevin dynamics on (latent) data space, such sampling is performed on the parameter space of inference model and is shown to converge to the true posterior under several mild conditions. The experiments demonstrate the generation quality of the proposed model.

**Summary Of The Review:**

The paper study an important problem, but the current presentation makes it hard to follow. Also the extra experiments and analysis need to be added to motivate and back-up their claims.

---

> ### Author Response · Authors · 2021-11-18
> **Response to Reviewer Lr3z**
>
> Thank you for your feedback.
> We answer your comment to address your concern.
> We also have some updates on our paper, so please read check it too.
>
> **What $u$ represents in Eq. (14)**
>
> $u$ is an input of the sampler function, which is defined in Section 3.3.
> We have added a statement in Section 4 to clarify it in the updated version.
>
> **How to choose $g$**
>
> In our experiment, we used a randomly initialized neural network for the feature extractor $g$ and freeze the weights during the training.
> We have added a statement to clarify it in Section 3.2.
> We found that it works well in practice, but we think that there could be a better way than random initialization.
> Developing a more sophisticated way is one possible future work.
> We also have added a statement about it in the conclusion.
>
> **Why Langevin dynamics is also used for the generator**
>
> In Section 4, we take a Bayesian approach, where the training is formulated as a calculation of the joint posterior $p \left( \Theta, Z \mid X \right)$.
> The reason we take the Bayesian approach here is that the learning algorithm can be more simply described than formulated as maximization of the marginal likelihood.
> In the updated version, we have added the learning algorithm based on maximum likelihood in Appendix B.
>
> **Baseline results should be improved**
>
> Based on your feedback, we have reimplemented the baseline models using the official implementation of Pang et al. (2020a), which is available at https://github.com/bpucla/latent-space-EBM-prior.
> By this update, the results of baseline models have been greatly improved, and we find that LEBM outperforms our LAE in SVHN and CelebA-HQ.
> As Reviewer vRp6 pointed out, it is natural that LEBM could perform better than our LAE, because LEBM conducts exact sampling from the posterior and the EBM, whereas LAE's amortized sampling may have some approximation error.
> Hence, the advantage of LAE over LEBM is still the efficiency of sampling by the power of amortization.
>
> About the FID score reported in Pang et al. (2020a), we are not sure that it is really convincing for several reasons.
> In fact, we ran the training for the SVHN dataset using their official code, but the result does not match the reported value (29.44).
> In addition, they did not report the error bar in their paper, so it may be a cherry-picked result.
> For these reasons, in the updated version, we reported the FID that we obtained by running the official code instead of the reported score in their paper.
>
> We would be glad to respond to any further questions and comments that you may have.
>
> Thanks.

---

> ### Author Response · Authors · 2021-11-26
> **A Gentle Reminder to Reviewer Lr3z**
>
> Thank you again to your efforts in reviewing our paper and your constructive comments. This is a gentle reminder that we have updated the paper to address your concerns. The discussion period will end soon, so please let us know if you have further comments about the update.
>
> Thanks!

---

### Official Review · Reviewer_siJZ · 2021-11-02

**Correctness:** 3
**Technical Novelty And Significance:** 3
**Empirical Novelty And Significance:** Not applicable
**Recommendation:** 6
**Confidence:** 3

**Main Review:**

Strengths:

(1) The idea of the paper is novel and interesting. Theoretical analysis is provided.

(2) The proposed method can be applied to sampling both the posterior distribution and the energy-based model.

(3) The proposed method is different from variational approximation.

Weaknesses:

(1) One thing I am concerned with is, if all the randomness is accounted for by the Langevin sampling of the parameters, will the generated samples be correlated?

(2) Can you always find a good sampler function to approximate complex target distribution?

**Summary Of The Paper:**

This paper proposes an amortized Langevin dynamics for sampling from a posterior distribution of a top-down generative model or from an energy-based model. The key is to recruit a sampler function that generates many samples directly in parallel, and run a single Langevin dynamics on the parameters of this sampler function. The method is illustrated by image generation.

**Summary Of The Review:**

The paper proposes a new idea on sampling from unnormalized densities. It can be useful for learning deep generative models.

---

> ### Author Response · Authors · 2021-11-18
> **Response to Reviewer siJZ**
>
> Thank you for your insightful feedback.
> We answer your comment to address your concern.
> We also have some updates on our paper, so please check it too.
>
> **Correlation of ALD samples**
>
> As you pointed out, the samples by ALD have correlations between data points, but it is not a problem for posterior approximation, because the true posterior distribution also has correlation through the shared likelihood function $p \left( x \mid z \right)$.
> For the unconditional case, the samples have correlations between multiple MCMC chains, so the samples of each chain may not converge to the target distribution.
> However, the total samples of all chains have the target distribution as a stationary distribution, which is more important in practice; hence we think that the correlation does not become problematic either in the unconditional case.
>
> **How to find a good sampler function**
>
> In the experiment, we simply use a randomly initialized neural network for the sampler function, which we find works well in practice.
> But we believe that there might be a more sophisticated way to choose a good function, which is a possible future work.
> We have added a statement about it in the conclusion.
>
>
> We would be glad to respond to any further questions and comments that you may have.
>
> Thanks.

---

> ### Author Response · Authors · 2021-11-26
> **A Gentle Reminder to Reviewer siJZ**
>
> Thank you again to your efforts in reviewing our paper and your constructive comments. This is a gentle reminder that we have updated the paper to address your concerns. The discussion period will end soon, so please let us know if you have further comments about the update.
>
> Thanks!

---

### Official Review · Reviewer_ZHba · 2021-11-06

**Correctness:** 3
**Technical Novelty And Significance:** 3
**Empirical Novelty And Significance:** 2
**Recommendation:** 6
**Confidence:** 3

**Main Review:**

It is not clear to me how to get the function g in the inference network. Is the function g obtained through some preprocessing? How does the quality of it affect the posterior approximation result? How do the authors get it in the experiment section? It will be better to provide more explanation and details about it, since it seems to play an important role in approximating the posterior as shown in Theorem 1.

The second condition for ALD to converge to the true distribution does not seem mild. Since the dataset size is typically very large, it is impractical to have such high dimension for the linear layer. Therefor the claim seems a bit misleading that “ALD has the target posterior as a stationary distribution with a mild assumption”.

How does the storage cost of the proposed method compared to LD without amortization and AVI? By looking at Algorithm 1, the storage cost of the proposed method seems much higher, since it has to save Z^1 to Z^n and each Z^i contains several samples.

For the experiments, it will be better to add the results of SGALD in Figures 1 and 4 to check the posterior estimation when using minibatch of data. Since SGALD is the one being used in LAE in practice.

On the synthetic distributions, the dimension of the linear layer is much larger than the dataset size (128 vs 3). However in practice the dataset size will be much larger than the dimension of the linear layer (as what the authors did for the experiment on image generation). It will be much convincing to simulate this scenario on synthetic data and verify that with small dimension ALD/SGALD can still approximate posterior well. It will be very helpful to further show how the posterior estimation changes with respect to the dimension of the linear layer.



**Summary Of The Paper:**

This paper introduces amortized Langevin dynamics for latent variables and extends it to unconditional distributions. A Langevin autoencoder is further developed by using amortized Langevin dynamics for prior and posterior sampling. The proposed method has been tested on synthetic distributions and image generation.

**Summary Of The Review:**

In summary, I think the idea is reasonable and developing a cheap MCMC-based autoencoder will be of interest to the community. However, I have some concerns with respect to the methodology and the experiments mentioned above.

---

> ### Author Response · Authors · 2021-11-18
> **Response to Reviewer ZHba**
>
> Thank you for your insightful comments.
> We answer your comment to address your concern.
> We also have some updates on our paper, so please check it too.
>
> **How to get the function $g$ in the inference network**
>
> We used a randomly initialized neural network for the feature extractor $g$ and freeze the weights throughout the training.
> We have added a statement to clarify it in Section 3.2.
> Although it works well in practice, we think that there could be a better way than random initialization.
> Developing a more sophisticated way is one possible future work.
> We also have added a statement about it in the conclusion.
>
> **Assumption of Theorem 1**
>
> We agree that the second condition of Theorem 1 is not so mild, so we have changed the expression from "with a mild assumption" to "under some assumptions".
>
> **Storage cost of ALD**
>
> The storage cost of ALD is the same as traditional LD because traditional LD also needs to store all samples per data point.
> AVI's storage cost may be cheaper than ALD because they only store its variational parameters.
> But when obtaining samples from the variational posterior, the cost is the same as ALD, i.e., $N_\mathrm{sample} \times N_\mathrm{data}$.
>
> **Toy examples of SGALD**
>
> The samples of SGALD are provided in Figure 2, although their posterior is univariate.
> > For the experiments, it will be better to add the results of SGALD in Figures 1 and 4 to check the posterior estimation when using minibatch of data. Since SGALD is the one being used in LAE in practice.
>
> Does this mean that we should also provide SGALD samples of multivariate posteriors to check if SGALD can capture the correlation among dimensions?
>
> **Effects of the dimensionality of inference models**
>
> We added an experiment to investigate how the dimensionality of the last linear layer affects the sample quality in Appendix E.
> When the dimensionality is too small, the sample quality deteriorates for some data points.
> This empirical result is identical to what our theorem suggests.
>
> We would be glad to respond to any further questions and comments that you may have.
>
> Thanks.

---

> ### Author Response · Authors · 2021-11-26
> **A Gentle Reminder to Reviewer ZHba**
>
> Thank you again to your efforts in reviewing our paper and your constructive comments. This is a gentle reminder that we have updated the paper to address your concerns. The discussion period will end soon, so please let us know if you have further comments about the update.
>
> Thanks!

---

### Author Response · Authors · 2021-11-18
**Overall Updates**

Thank you very much for your useful comments on our paper.
Based on the feedback we received, we have made some updates to our paper.
The updated parts are shown in red text.

Below is a summary of the major changes.

**Updates on empirical results of image generation**

Some reviewers have concerns about our empirical results due to the poor performance of the baseline models.
To improve it, we re-implemented the baseline models using the official implementation of Pang et al. (2020a), which is available at https://github.com/bpucla/latent-space-EBM-prior.
By this update, the results of baseline models have been greatly improved.

**Additional experiment on amortization gap**

To investigate the effect of the capacity of the inference model, we have added a toy experiment in Appendix E.
We used the same experimental settings with the 2d toy example in Section 6.1, and change the dimensionality of the last linear layer of the inference model in a range from 2 to 128.
The result shows that when the dimension is smaller than the number of data points, the sample quality deteriorates for some data points.
This is identical to what Theorem 1 suggests.

**Other minor revisions**

We have added or changed some expressions of our writing to clarify our motivation of the paper.

Thanks.

---

### Decision · Program_Chairs · 2022-01-20

**Decision:**

Reject

**Comment:**

This paper proposes an amortization strategy for MC sampling from a single chain rather than per-datapoint chains, and uses this strategy to define a new Bayesian autoencoder based on Langevin dynamics.

The reviewers find the line of thought very promising, and a potentially interesting addition to the latent variable literature, while also raising some concerns. The dimension of the single chain must match the dataset size, which limits the computational benefits coming from amortization, and in fact this restriction seems hard, as empirical results (added in the discussion period) are qualitatively worse in the `d<n` case. This could be emphasized much more strongly in the current version, and seems worth deeper investigation. In the discussion, the authors agreed that in the case when the feature matrix G is the identity matrix then there can be no amortization improvement, but for other choices of (fixed) features, amortization *can* yield improvements; this is quite unclear. In addition, in response to the reviewers' observation, the authors improved in the discussion period the implementation of the EBM baseline, leading to much less clear cut differences on metrics. To improve the work further, the authors should clarify the source of amortization improvement, and discuss more the relationship to Bayesian Neural Networks (perhaps by evaluating against Bayesian / hyper-net / hyper-GAN generative models.)